# Gradient Smoothing: Coupling Layer-wise Updates for Improved Optimization

Haoming Meng [* 1 2]   Anton Sugolov [* 1 2]   Vardan Papyan [1 2]

## Abstract

Deep neural networks with repeated blocks, such as transformers and ResNets, often exhibit structured relationships across layers that emerge during training. Motivated by this observation, we introduce a general paradigm of *Depth-wise Gradient Augmentation*, in which the update applied to a layer may depend on the base optimizer updates computed for other layers. We study an instantiation of this idea, termed *Gradient Smoothing*, which couples optimizer updates across depth and admits a natural interpretation as a structured preconditioning method. Our framework operates directly on block-wise update vectors produced by arbitrary base optimizers (e.g., SGD, Adam, Muon), applying structured depth-wise smoothing operators such as local weighted averaging with minimal computational overhead. We evaluate Gradient Smoothing across a diverse set of architectures and training regimes, including language model pretraining, RL post-training of LLMs on reasoning tasks, diffusion modeling, and image classification with Vision Transformers. Across these settings, Gradient Smoothing consistently improves convergence and generalization performance without modifying model architectures or training objectives. We further show that smoothing promotes more structured representation evolution across depth, suggesting a connection between structured update coupling and the internal organization of learned representations. These results position Gradient Smoothing as a simple and broadly applicable approach for improving training in modern deep networks.

---

[*]Equal contribution  [1]Department of Mathematics, University of Toronto, Toronto, Canada  [2]Vector Institute, Toronto, Canada.  Correspondence to: Haoming Meng <haoming.meng@mail.utoronto.ca>.

*Proceedings of the 43rd International Conference on Machine Learning*, Seoul, South Korea. PMLR 306, 2026. Copyright 2026 by the author(s).

## 1. Introduction

Many modern deep neural networks are composed of repeated architectural blocks with shared computational structure, such as Residual Networks (ResNets) (He et al., 2015) and transformers (Vaswani et al., 2023). Recent work has shown that networks of this form often exhibit strong structural regularities across depth that emerge during training. In particular, studies on *Transformer Block Coupling* (Aubry et al., 2025) and *Residual Alignment* (Li & Papyan, 2024) demonstrate that singular vectors of block Jacobians and residual representations become aligned across layers, suggesting a form of implicit coordination throughout depth. Related evidence appears in studies of layer redundancy and pruning in large language models (Gromov et al., 2025), as well as analyses showing smooth evolution of information-theoretic quantities across depth (Skean et al., 2024).

These observations suggest that repeated blocks in deep networks do not evolve independently, but instead exhibit coordinated representational and dynamical structure across depth. However, standard optimization methods typically treat block-wise optimizer updates independently (aside from their dependencies through the forward and backward passes), applying updates separately to each layer.

This mismatch between architectural structure and optimization dynamics raises a natural question: *should optimizer updates across depth themselves be coupled?*

In this work, we introduce a general paradigm of *Depth-Wise Update Coupling*. Rather than applying the update produced for each block in isolation, the update applied to a layer may depend on updates computed for other layers. We study a simple and efficient instantiation of this idea, termed *Gradient Smoothing*, which applies structured depth-wise smoothing operators to block-wise update vectors.

Gradient Smoothing is simple to implement, incurs minimal computational overhead, and is compatible with arbitrary base optimizers. Moreover, the framework admits a natural interpretation as a structured preconditioning method acting on block-structured parameter spaces, enabling both theoretical analysis and principled design of update coupling operators.

**Contributions.**

1. We propose *Gradient Smoothing*, a simple and optimizer-agnostic framework of a broader paradigm of *Depth-wise Gradient Augmentation*, in which gradient updates are coupled across transformer blocks using depth-wise operators acting on the gradient updates. Our framework applies structured depth-wise smoothing operators to block-wise update vectors with minimal computational overhead and is compatible with arbitrary base optimizers.

2. We demonstrate that Gradient Smoothing consistently improves optimization and generalization performance across a diverse set of architectures and training regimes, including language model pretraining, RL post-training of LLMs for reasoning, supervised learning with Vision Transformers, and diffusion models.

3. We provide empirical evidence that Gradient Smoothing promotes more structured representation evolution across depth, as measured through layer-wise trajectory alignment and representation similarity analyses.

4. We provide theoretical analysis of Gradient Smoothing and its effect on and representational structure, with results supported by empirical observations.

## 2. Background and Setup

### 2.1. Block-Structured Optimization

We consider neural networks composed of $L$ repeated blocks connected by skip connections, where each block shares the same functional form but has its own parameters. In addition, the model may include a (typically small) collection of parameters not associated with the repeated structure, such as input embeddings or output heads.

Formally, let $F : \mathbb{R}^d \times \mathbb{R}^p \to \mathbb{R}^d$ denote the residual mapping associated with a single block. For block $\ell$, the residual function is given by $F(\cdot; \theta_\ell)$, where $\theta_\ell \in \mathbb{R}^p$ are layer-specific parameters. Given the initial embedding $h_0$, the hidden states $(h_\ell)_{\ell=1}^L \subset \mathbb{R}^d$ then evolve according to

$$h_{\ell+1} = h_\ell + F(h_\ell; \theta_\ell), \qquad \ell = 1, \ldots, L-1. \quad (1)$$

We write the full parameter vector as

$$(\theta, \phi), \qquad \theta = (\theta_1, \ldots, \theta_L), \quad \theta_l \in \mathbb{R}^p,$$

where $\phi$ collects all non-block parameters. This setting encompasses a broad class of modern architectures, including residual networks with repeated block structure, Vision Transformers, and Transformer-based language models (by letting each $h_\ell$ be a sequence of vectors). We denote the training objective by $\mathcal{L}(\theta, \phi)$ and assume it is differentiable.

At optimization step $t$, the block-wise gradients are given by

$$g_l^{(t)} := \nabla_{\theta_l} \mathcal{L}(\theta^{(t)}, \phi^{(t)}), \qquad g^{(t)} := (g_1^{(t)}, \ldots, g_L^{(t)}),$$

and we denote by $g_\phi^{(t)} := \nabla_\phi \mathcal{L}(\theta^{(t)}, \phi^{(t)})$ the gradient with respect to $\phi$.

Standard first-order methods treat the block-wise gradients $\{g_l^{(t)}\}_{l=1}^L$ independently. However, repeated-block architectures induce strong structural relationships across depth: adjacent blocks often learn similar transformations and exhibit aligned representations, particularly after an initial transient phase of training. This suggests that the stacked gradient vector $g^{(t)}$ possesses meaningful structure along the depth dimension.

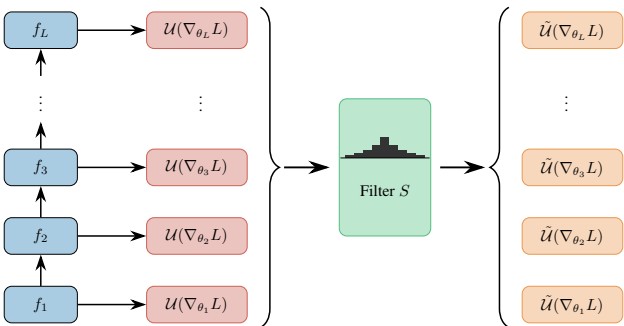

*Figure 1.* **Gradient Smoothing diagram.** Representation of the gradient smoothing scheme applied across depth to the gradients of a deep network after backpropagation. For a deep network with $L$ identical layers (but with different parameters), the gradient updates with respect to in each parameter $\theta^l$ are reweighted across depth to stabilize information propagation.

### 2.2. General First-Order Updates

Rather than committing to a specific optimizer, we consider a general first-order update rule of the form

$$(\theta^{(t+1)}, \phi^{(t+1)}) = (\theta^{(t)}, \phi^{(t)}) - \eta \, (u^{(t)}, u_\phi^{(t)}),$$

where

$$u^{(t)} := \mathcal{U}^{(t)}(g^{(t)}), \qquad u_\phi^{(t)} := \mathcal{U}_\phi^{(t)}(g_\phi^{(t)})$$

denote the updates produced by the base optimizer for the block and non-block parameters, respectively.

This formulation includes gradient descent ($u^{(t)} = g^{(t)}$), momentum methods, adaptive optimizers such as Adam or AdamW, as well as Muon and related variants. Importantly, $u^{(t)}$ may depend on both the current gradient and internal optimizer state, and may involve nonlinear or time-varying transformations of $g^{(t)}$.

This optimizer-agnostic viewpoint allows us to study update augmentation directly at the level of optimizer updates, independent of the specific mechanisms used to construct them.

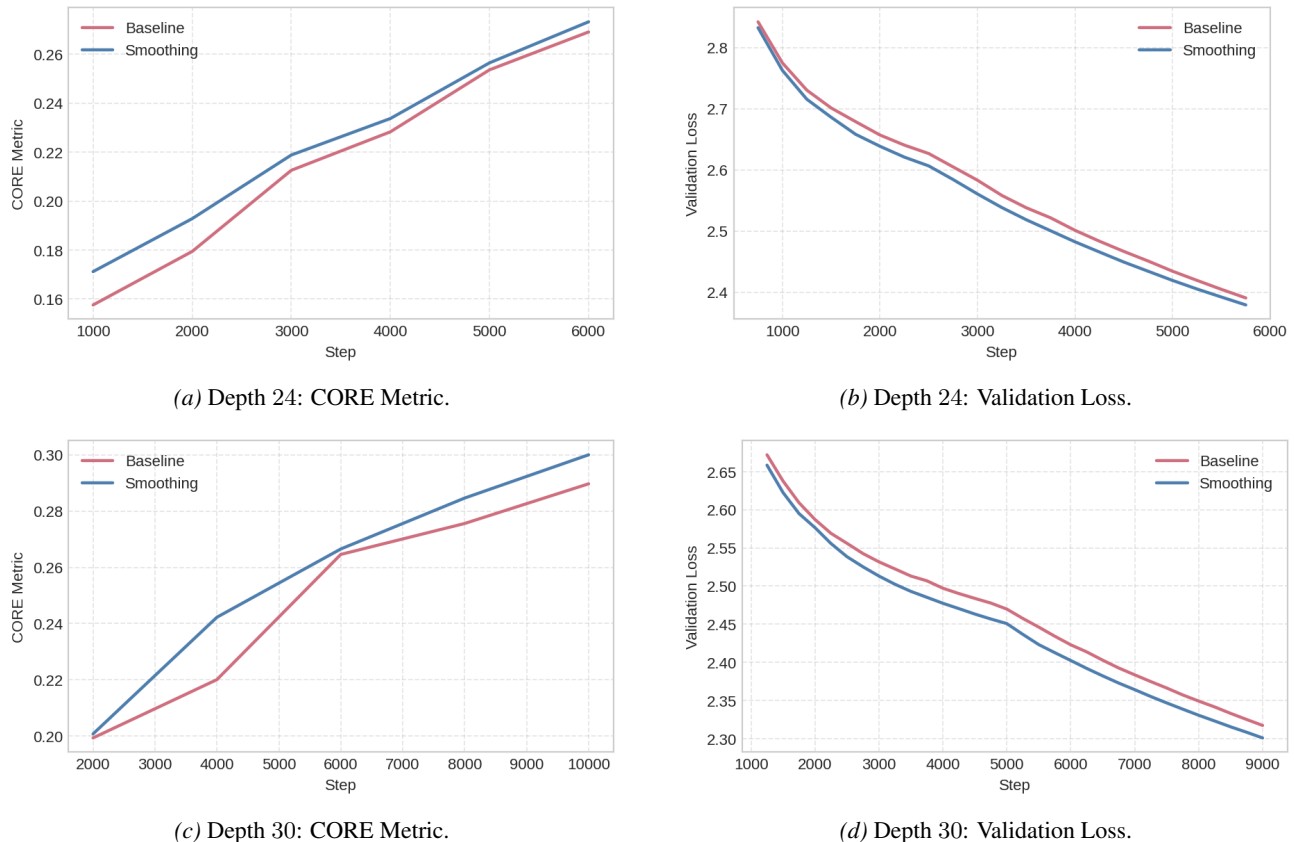

*(a)* Depth 24: CORE Metric.

*(b)* Depth 24: Validation Loss.

*(c)* Depth 30: CORE Metric.

*(d)* Depth 30: Validation Loss.

*Figure 2.* **Nanochat pretraining with Gradient Smoothing.** Validation loss and DCLM CORE metric throughout `nanochat` pretraining of GPT2 under the default Adam + NorMuon optimization setup, comparing baseline against an otherwise identical run with depth-wise update smoothing applied to the repeated transformer blocks. Top row: depth 24 with $\alpha = 0.05$. Bottom row: depth 30 with $\alpha = 0.1$. At both depths, smoothing accelerates convergence of validation loss and improves the CORE metric throughout training, with a larger gap visible at depth 30.

## 3. Gradient Smoothing Framework

### 3.1. Depth-wise Gradient Augmentation

Networks built from repeated blocks develop strong cross-depth structure in both their representations and their optimization dynamics. Motivated by this, we study depth-wise gradient augmentation, a general paradigm in which the optimizer update applied to a layer may depend on updates computed for other layers. We develop the method in general before presenting a simple and effective augmentation method, *window smoothing*, where gradients are updated by averaging a window of adjacent layers.

Let $\theta = (\theta_1, \ldots, \theta_L)$ denote the parameters of the repeated blocks of a network, where each $\theta_l \in \mathbb{R}^p$, and let $\phi$ denote the remaining (non-block) parameters. Writing $D_\theta = Lp$, the repeated-block parameters lie in $\mathbb{R}^{D_\theta}$, while the non-block parameters lie in $\mathbb{R}^{D_\phi}$.

At iteration $t$, let

$$u^{(t)} = (u_1^{(t)}, \ldots, u_L^{(t)}) \in \mathbb{R}^{D_\theta}$$

denote the block-wise update produced by a base optimizer (e.g., SGD, Adam, Muon), together with the corresponding update $u_\phi^{(t)}$ for the non-block parameters.

In its most general form, Depth-wise Gradient Augmentation constructs transformed updates

$$\tilde{u}_\ell^{(t)} = \mathcal{C}_\ell\big(u_1^{(t)}, \ldots, u_L^{(t)}\big),$$

where each $\mathcal{C}_\ell$ specifies how updates across depth are combined to produce the update applied to layer $\ell$.

In this work, we study a simple instantiation of this framework, which we term *Gradient Smoothing*. We introduce a linear operator $S \in \mathbb{R}^{L \times L}$ acting on the layer index, and define the block-structured operator

$$P := S \otimes I \in \mathbb{R}^{D_\theta \times D_\theta},$$

where $I$ denotes the identity operator on the within-block parameter space $\mathbb{R}^p$.

The transformed block-wise update is then given by

$$\tilde{u}^{(t)} := P u^{(t)} = (S \otimes I) u^{(t)}.$$

To describe the update on the full parameter vector $(\theta, \phi)$, we lift $P$ to the block-diagonal operator

$$\bar{P} := \begin{pmatrix} P & 0 \\ 0 & I_\phi \end{pmatrix} \in \mathbb{R}^{(D_\theta + D_\phi) \times (D_\theta + D_\phi)},$$

where $I_\phi$ denotes the identity operator on the $\phi$-coordinates.

Given the full optimizer update

$$u_{\text{full}}^{(t)} := \left( u^{(t)}, u_\phi^{(t)} \right),$$

Gradient Smoothing applies $\bar{P}$ to obtain

$$\tilde{u}_{\text{full}}^{(t)} = \bar{P} u_{\text{full}}^{(t)} = \left( P u^{(t)}, u_\phi^{(t)} \right).$$

The resulting optimization step becomes

$$\theta^{(t+1)} = \theta^{(t)} - \eta\, \tilde{u}^{(t)}, \qquad \phi^{(t+1)} = \phi^{(t)} - \eta\, u_\phi^{(t)}.$$

Gradient Smoothing thus corresponds to a structured depth-wise coupling of optimizer updates, or equivalently, a structured preconditioning operator acting on the block-wise update vector. Importantly, this construction is independent of the specific choice of base optimizer and operates purely at the level of update vectors.

### 3.2. Smoothing Operators

To consider a simple example of Gradient Smoothing, take $S$ to be the local window averaging operator across depth. Let $u = (u_1, \ldots, u_L)$ denote the stacked block-wise update for $\theta$; non-block parameters $\phi$ are updated by the base optimizer and are not smoothed. The smoothed update is given by

$$\tilde{u}_l = \begin{cases} \left(1 - \frac{\alpha}{2}\right)u_1 + \frac{\alpha}{2}u_2, & l = 1, \\ (1 - \alpha)u_l + \frac{\alpha}{2}(u_{l-1} + u_{l+1}), & 2 \le l \le L - 1, \\ \left(1 - \frac{\alpha}{2}\right)u_L + \frac{\alpha}{2}u_{L-1}, & l = L, \end{cases}$$

(2)

where $\alpha \in [0, 1)$ controls the smoothing strength. Each block's update is replaced by a convex combination of its own update and those of its immediate neighbors in depth; the boundary cases at $l = 1$ and $l = L$ use a half-weight on the single available neighbor so that $S$ remains row-stochastic ($S\mathbf{1} = \mathbf{1}$). The resulting $S \in \mathbb{R}^{L \times L}$ is symmetric and introduces no additional trainable parameters.

Many other depth-wise weighting schemes are possible within the framework. For instance, exponential smoothing, higher-order tridiagonal kernels, or local window averaging with larger window sizes all fit the same general paradigm. Our work focuses on local window averaging as a simple yet effective instantiation of Gradient Smoothing, applicable to many optimizers and across broad tasks.

## 4. Experiments

We evaluate Gradient Smoothing across four distinct training regimes that stress different aspects of optimization: (i) RL fine-tuning of LLMs on reasoning tasks, (ii) language model pretraining, (iii) supervised image classification with Vision Transformers, and (iv) diffusion model training. To demonstrate compatibility with existing training setups, in all settings we compare against standard tuned baselines while keeping the underlying training *hyperparameters unchanged when applying smoothing*. As a result, the reported improvements are obtained without additional hyperparameter tuning specific to smoothing, suggesting that further gains may be possible with dedicated tuning. Because the method operates solely on parameter updates, it requires no architectural modifications and introduces negligible computational overhead, making it easy to integrate across a wide range of models and training regimes.

### 4.1. Experimental Setup

**RLVR for LLM Reasoning:** The base model we use is `DeepSeek-R1-Distill-Qwen-1.5B` (Guo et al., 2025; Yang et al., 2024). We train with Group Relative Policy Optimization (GRPO) (Shao et al., 2024) on the dataset from Open-RS (Dang & Ngo, 2026) (without the cosine response length reward), and then evaluate on downstream mathematical reasoning tasks: AIME24, AIME25, AMC23, MATH-500. We train the model for 200 optimization steps with AdamW using a base learning rate of $10^{-6}$ with a cosine learning-rate schedule and a minimum learning-rate ratio of $0.1$, including a linear warmup over the first $10\%$ of training steps. Evaluation sampling is done with temperature $0.6$ and top_p $0.95$.

**LLM Pretraining:** We further evaluate gradient smoothing in a language-model pretraining setting using the `nanochat` recipe (Karpathy, 2025). `nanochat` provides a compact end-to-end GPT-style pretraining pipeline in which the transformer depth is the main scale parameter, while model width, number of heads, learning-rate scaling, training horizon, batch size, and weight decay are automatically determined from this depth. We train decoder-only transformer models with depths 24 and 30 using the default `nanochat` architecture and Muon-style (Jordan et al., 2024) optimization setup with NorMuon (Li et al., 2025). For each depth, we compare the baseline run with an otherwise identical run where depth-wise update smoothing is applied to the repeated transformer blocks. We keep all other training settings fixed and evaluate using validation loss and the DCLM CORE metric from the `nanochat` evaluation suite.

**Vision:** ViT-B/4 models following DeiT training (Touvron et al., 2021) with default image augmentations. We train

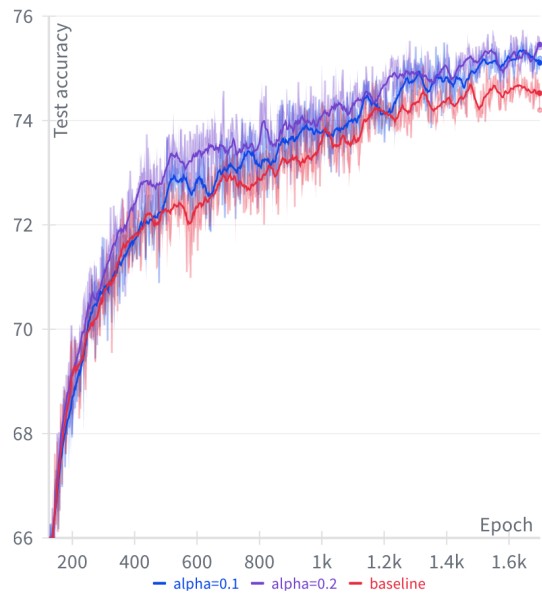

*Figure 3.* **Test accuracy improvement with smoothing.** Test accuracy of ViT-B trained on CIFAR-100 with DeiT training recipe with data augmentations for 1700 epochs. We compare baseline training ($\alpha = 0$, red) against window smoothing with $\alpha = 0.1$ (blue) and $\alpha = 0.2$ (purple). Solid lines show running average trendlines (25 epochs) for clarity. Both smoothing configurations outperform baseline, with $\alpha = 0.2$ achieving a final test accuracy of 75.62% compared to the baseline 74.56% (Table 2), and suggests that moderate gradient smoothing provides a consistent performance benefit throughout training.

for 1700 epochs using AdamW with a cosine learning rate schedule, initial learning rate of $5 \times 10^{-4}$, and weight decay of 0.05. CIFAR-100 evaluated via top-1 accuracy.

**Diffusion:** We evaluate gradient smoothing on U-ViT (Bao et al., 2023), a ViT-based diffusion backbone with long skip connections proposed for image generation on CIFAR-10. The U-ViT architecture has 12 transformer blocks with embedding dimension 512 and 8 attention heads. Optimization uses AdamW with learning rate $2 \times 10^{-4}$, weight decay 0.03, batch size 128, with a 2500-step warmup for 500,000 steps. We sample via Euler–Maruyama SDE at 50 NFE and report FID against both 10k and 50k generated samples.

**Gradient Smoothing Configuration.** Gradient smoothing is applied across block-wise updates at each optimization step (Section 3.2) with $\alpha$-window averaging across depth. In order to stabilize the effect of smoothing, we also investigate two normalization settings;

i) **Rescaled Smoothing.** Updates $u_l$ are rescaled to their original norm after being smoothed

$$\tilde{u}_\ell = \frac{\|u_\ell\|_2}{\|(Pu)_\ell\|_2}(Pu)_\ell.$$

ii) **Directional Smoothing.** Updates $u_l$ are rescaled to unit norm, $u_l/\|u_l\|_2$, before applying smoothing

$$\tilde{u}_\ell = \frac{\|u_\ell\|_2 \, (P(u_1/\|u_1\|_2, \ldots, u_L/\|u_L\|_2))_\ell}{\|(P(u_1/\|u_1\|_2, \ldots, u_L/\|u_L\|_2))_\ell\|}.$$

In addition to experimenting with $\alpha$-window settings, we smooth over standard, rescale, and directional stabilizations for update smoothing. We also consider varied parameter groups within each block that are affected by any update smoothing operations. In Transformers/ViTs, most capacity and gradient signal flows through linear projections (attention $W_Q, W_K, W_V, W_O$ and MLP layers), while normalization and embedding parameters are lower-dimensional and often behave differently under optimization. Specifically we distinguish between (i) smoothing applied to *all* block parameters (full), (ii) smoothing applied *only* to linear layers within each block (the non-normalization layers in the models we use) (proj).

### 4.2. Experimental Results

Gradient smoothing consistently improves performance across the diverse set of training regimes and benchmarks we evaluate. Notably, even simple local window smoothing is effective across domains, suggesting that the benefits do not depend on carefully engineered or highly specialized smoothing operators, though more sophisticated operators may yield further gains. Importantly, in all experiments, smoothing is applied directly on top of already tuned baseline training setups without modifying the underlying training hyperparameters, and still yields consistent improvements despite no additional hyperparameter tuning being performed specifically for smoothing.

**RLVR Fine-Tuning for LLM Reasoning:** Gradient smoothing improves the reasoning accuracy of `DeepSeek-R1-Distill-Qwen-1.5B` after RLVR fine-tuning across mathematical benchmarks (Table 1). Window smoothing with $\alpha = 0.1$ achieves a substantial improvement over the baseline (57.71% vs. 53.03%), with particularly strong gains on AIME24 (43.33% vs. 30.00%) and AIME25 (30.00% vs. 23.33%). Notably, these improvements are obtained without modifying the RL objective or reward structure, suggesting that even regularizing just the model's internal structure can improve and stabilize RL-based reasoning fine-tuning.

**LLM Pretraining:** Gradient smoothing also improves optimization dynamics in language-model pretraining using

*Table 1.* LLM mathematical reasoning performance (`pass@1` accuracy) after RLVR fine-tuning. Gradient Smoothing improves accuracy substantially on downstream mathematical benchmarks compared to fine-tuning regularly with AdamW.

| Method | AIME24 | AIME25 | AMC23 | MATH-500 | Avg |
|---|---|---|---|---|---|
| Base model | 26.67 | 16.67 | 70.00 | 83.00 | 49.09 |
| Baseline (AdamW) | 30.00 | 23.33 | 75.00 | 83.80 | 53.03 |
| + Window ($\alpha = 0.1$, Rescale, Proj) | 36.67 | **30.00** | **77.50** | **85.44** | 57.39 |
| + Window ($\alpha = 0.1$, Rescale, Full) | **43.33** | **30.00** | 72.50 | 85.00 | **57.71** |

*Table 2.* Image classification accuracy (%) for ViT-B on CIFAR-100.

| Method | CIFAR-100 |
|---|---|
| Baseline (AdamW) | 74.56 |
| + Window ($\alpha = 0.1$, Rescale, Proj) | 75.44 |
| + Window ($\alpha = 0.2$, Directional, Proj) | **75.62** |

*Table 3.* FID scores on CIFAR-10 for U-ViT diffusion training with and without Gradient Smoothing.

| Method | FID@10k | FID@50k |
|---|---|---|
| Baseline (AdamW) | 6.58 | 4.01 |
| + Window ($\alpha = 0.2$, Rescale, Proj) | **5.82** | **3.74** |

the `nanochat` recipe (Figure 2). Across both depth-24 and depth-30 models, smoothing consistently accelerates convergence of the validation loss while simultaneously improving the CORE metric throughout training. The improvements emerge early in optimization and persist across training trajectory, suggesting that smoothing provides a stable inductive bias for the model. Notably, these gains are obtained in a highly optimized pretraining setup using the mixed NorMuon/AdamW optimization pipeline from `nanochat`, indicating that the benefits of smoothing extend beyond standard Adam-family training alone. The stronger improvements observed for the deeper depth-30 model further suggest that coupling updates across depth may become increasingly beneficial as the model depth increases.

**Vision Transformer Image Classification:** Gradient smoothing improves test accuracy of ViT-B on CIFAR-100 (Table 2, Figure 3). Window smoothing with $\alpha = 0.2$ achieves notable improvements (75.62% vs 74.56% baseline), while $\alpha = 0.1$ also improves over baseline (75.44%). These results suggest that gradient smoothing provides consistent regularization for vision transformer training, with gains emerging early and persisting throughout optimization.

**Diffusion:** We evaluate gradient smoothing on U-ViT (Bao et al., 2023), a ViT-based diffusion backbone which provides a strong tuned baseline on CIFAR-10. With 50 NFE sampling, window smoothing (without smoothing LayerNorm parameters) reduces FID@10k from 6.58 to 5.82 and FID@50k from 4.01 to 3.74 (Table 3). All runs share identical architecture, hyperparameters, and sampling budget, so the gains are not attributable to additional tuning in the smoothed setting. These results suggest that gradi-

ent smoothing additionally extends to transformer-based generative modeling for images.

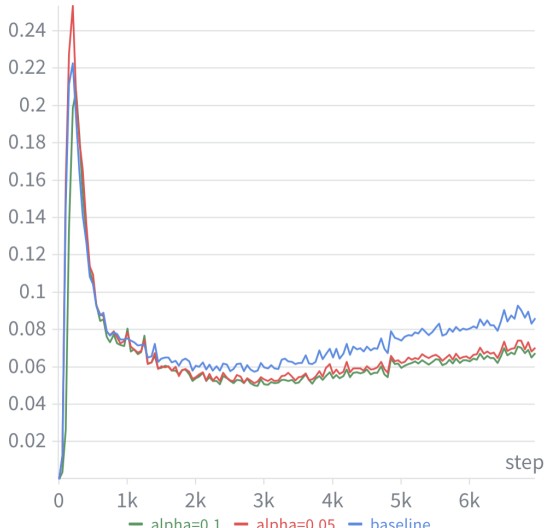

*Figure 4.* **Microbatch gradient variance with window smoothing.** Total microbatch variance $V_{\mathrm{mb}}$ (Section 4.4) during nanochat depth 24 pretraining, comparing the baseline (blue) against window smoothing with $\alpha = 0.05$ (red) and $\alpha = 0.1$ (green). After the initial warmup, both smoothing runs maintain consistently lower microbatch gradient variance than baseline, with the gap widening later in training.

### 4.3. Effect of Smoothing on Representations

Smoothing parameter updates suggests that at convergence, models may be tend towards more coordinated transformations across depth. To quantify this effect on learned representations, we measure two geometric properties of token trajectories: (i) cosine similarity between successive

residual updates, and (ii) line shape score (LSS), which captures how close the trajectory lies to a one-dimensional affine space.

**Cosine similarity of residual.** To quantify the change in representation induced by a layer, we consider the differences in token representation across depth $x_\ell$, for $1 \leq \ell \leq L$. For $\ell \in \{1, \ldots, L-1\}$ the difference $d_\ell = x_{\ell+1} - x_\ell$ corresponds to the contribution of layer $\ell + 1$. The similarity of contributions between layers is quantified by $c_\ell = \frac{\langle d_\ell, d_{\ell+1}\rangle}{\|d_\ell\|\|d_{\ell+1}\|}$ (Figure 6).

**Line shape score.** We measure the linearity of token trajectories via the line shape score (LSS) (Gai & Zhang, 2021), measuring the closeness of a trajectory to a line. Given representations $x_0, \ldots, x_L$, the normalized trajectory $\tilde{x}_0 = x_0$ with

$$\tilde{x}_\ell = \tilde{x}_{\ell-1} + \frac{x_\ell - x_{\ell-1}}{\|x_\ell - x_{\ell-1}\|_2}$$

The line shape score is the given by LSS $= \frac{L}{\|\tilde{x}_L - \tilde{x}_0\|_2}$. LSS $\geq 1$, with equality if and only if the trajectory is perfectly co-linear (Figure 7).

We measure the effect of smoothing on representation structure via the Line Shape Score (LSS) and cosine similarity of CLS token trajectories for ViT-B trained with window smoothing for $\alpha \in \{0.1, \ldots, 0.4\}$ relative to the non-smoothed baseline ($\alpha = 0.0$). As the smoothing strength increases, layer-wise contributions to the CLS token become more aligned (Figure 5), and the trend persists when measuring mean cosine similarity of the contribution at individual layers (Figure 6). Smoothing also increases the linearity of the CLS token trajectory across layers (Figure 7). Together, these results suggest that gradient smoothing may implicitly regularize representation paths, encouraging more coherent evolution of the CLS token through the model depth.

### 4.4. Effect of Smoothing on Training Stability

**Gradient variance.** To assess optimization stability, we measure the total stochastic gradient variance across microbatches at each logging step:

$$V_{\text{mb}} = \frac{1}{N} \sum_{n=1}^{N} \|g^{(n)} - \bar{g}\|_2^2,$$

where $g^{(n)}$ is the stacked per-microbatch gradient and $\bar{g}$ its sample mean over the $N$ batch elements at that step. Smoothing runs with $\alpha \in \{0.05, 0.1\}$ maintain consistently lower variance than baseline throughout nanochat pretraining (Figure 4), suggestive of improved training stability. Full measurement details are deferred to Appendix A.5, and a variance contraction bound is shown across layer depth (Proposition A.10).

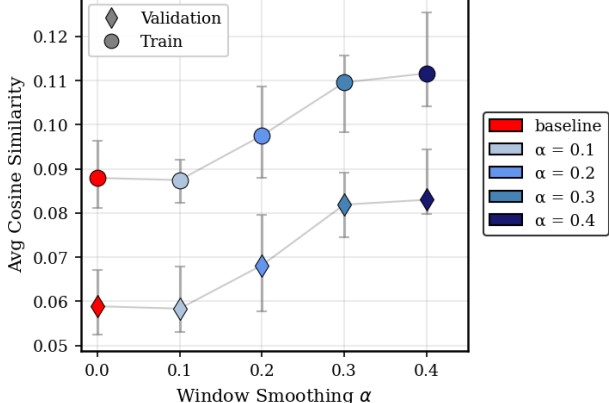

*Figure 5.* **Layer contributions are similar with increased smoothing.** Average cosine similarity of layer differences $d_\ell = x_{\ell+1} - x_\ell$ for ViT-B CLS token trained on CIFAR-100 (1700 epochs). Means are taken across all layer pairs $(d_i, d_j)$ for $1 \leq i, j \leq L-1, i \neq j$. Each point shows the median across 100 images from the validation set (rhombus) and training set (circle); error bars indicate interquartile range (Q1–Q3). As the window smoothing parameter $\alpha \in \{0.1, \ldots, 0.4\}$ increases from the baseline ($\alpha = 0$), mean cosine similarity increases monotonically, suggesting that gradient smoothing encourages more consistent directional updates across layers.

## 5. Alignment of Representation Differences under Gradient Smoothing

This section formalizes a consistent empirical observation: *gradient smoothing increases the alignment of representation differences across depth.* We study this through the residual increments $r_\ell := h_{\ell+1} - h_\ell$, and the cosine similarity $\cos(r_{\ell+1}, r_\ell)$, which measures how much consecutive residual increments point in the same direction. Our analysis gives one-step lower bounds on the *average cosine* after applying either the baseline optimizer update or its smoothed version. The key mechanism is that local-averaging smoothing contracts depth-to-depth variation in the update directions.

**Residual increments and cosine metric.** For any depth-stacked parameters $\vartheta = (\vartheta_1, \ldots, \vartheta_L)$, define for $\ell = 1, \ldots, L$,

$$r_\ell(\vartheta) := h_{\ell+1}(\vartheta) - h_\ell(\vartheta) = F\big(h_\ell(\vartheta); \vartheta_\ell\big) \in \mathbb{R}^d. \quad (3)$$

We measure directional alignment of consecutive residual increments by

$$\overline{\cos}(\vartheta) := \frac{1}{L-1} \sum_{\ell=1}^{L-1} \cos\big(r_{\ell+1}(\vartheta), r_\ell(\vartheta)\big). \quad (4)$$

This metric is invariant to the magnitudes $\|r_\ell\|$ and isolates the alignment of representation *directions*. Empirically, we consistently observe that $\overline{\cos}$ increases under smoothing, as shown in Figure 5.

**One-step baseline and smoothed updates.** Let $u(\theta) = (u_1(\theta), \ldots, u_L(\theta))$ denote the blockwise update direction produced by a base optimizer. From current parameters $\theta$, define the baseline and smoothed one-step updates

$$\theta^b = \theta - \eta\, u(\theta), \qquad \theta^s = \theta - \eta\, Su(\theta), \qquad (5)$$

where $S$ is a depth-smoothing operator. We focus on the local-averaging smoother with $\alpha \in (0, \frac{1}{2}]$:

$$(Su)_\ell = \begin{cases} \left(1 - \frac{\alpha}{2}\right)u_1 + \frac{\alpha}{2}u_2, & \ell = 1, \\ (1 - \alpha)u_\ell + \frac{\alpha}{2}(u_{\ell-1} + u_{\ell+1}), & 2 \le \ell \le L - 1, \\ \left(1 - \frac{\alpha}{2}\right)u_L + \frac{\alpha}{2}u_{L-1}, & \ell = L. \end{cases}$$
$$(6)$$

**Assumptions.** We assume standard Lipschitz regularity of the block map $F$ and a mild non-degeneracy condition requiring the residual increments to have norms bounded away from zero along the relevant one-step trajectories. The formal assumptions are given in Appendix A.2.

### 5.1. Main Result

We now state the main one-step comparison. Unlike a simplified depth-flat analysis, this result allows the current parameters to already have nontrivial depth-to-depth variation. The comparison therefore separates two effects: the pre-existing depth variation $D\theta$ and the update-induced variation $Du(\theta)$.

**Theorem 5.1** (General one-step alignment comparison). *Assume the regularity conditions of Appendix A.2 (Assumption A.1). Let $S$ be the local-averaging smoother* (6) *with $\alpha \in (0, \frac{1}{2}]$, and let $D$ be the first-difference operator $(Dv)_\ell = v_{\ell+1} - v_\ell$. Define*

$$\delta := D\theta, \qquad w := Du(\theta),$$

*where norms and inner products are taken over the corresponding depth-stacked block vectors. Let $S^{(1)}$ denote the induced averaging operator on first differences, so that*

$$D(Su) = S^{(1)}Du$$

*as shown in Appendix A.2, Lemma A.5.*

*Then the smoothed and baseline one-step updates satisfy*

$$\overline{\cos}(\theta^s) \ge 1 - \frac{4L_h^2 M^2}{m^2} - \frac{4L_\theta^2}{m^2} \cdot \frac{1}{L-1} \left\| \delta - \eta S^{(1)}w \right\|^2,$$
$$(7)$$

$$\overline{\cos}(\theta^b) \ge 1 - \frac{4L_h^2 M^2}{m^2} - \frac{4L_\theta^2}{m^2} \cdot \frac{1}{L-1} \left\| \delta - \eta w \right\|^2. \quad (8)$$

*Moreover, the difference between these two lower bounds is exactly*

$$\frac{4L_\theta^2}{m^2} \cdot \frac{1}{L-1} \left[ \eta^2 \left( \|w\|^2 - \|S^{(1)}w\|^2 \right) - 2\eta\langle \delta, (I - S^{(1)})w\rangle \right].$$
$$(9)$$

*Consequently, if $\eta > 0$ and $L_\theta > 0$, smoothing gives a strictly improved lower bound whenever*

$$\langle \delta, (I - S^{(1)})w\rangle < \frac{\eta}{2}\left( \|w\|^2 - \|S^{(1)}w\|^2 \right). \qquad (10)$$

*In particular, since $S^{(1)}$ is a contraction with*

$$\|S^{(1)}\|_{\mathrm{op}} = \mu_\star := 1 - \alpha\left(1 - \cos\left(\frac{\pi}{L}\right)\right) < 1, \qquad (11)$$

*the update-induced depth variation satisfies*

$$\|S^{(1)}w\|^2 \le \mu_\star^2 \|w\|^2. \qquad (12)$$

The proof is given in Appendix A.2. A depth-flat special case, where $D\theta = 0$ and the improvement comes purely from contraction of the update differences, is discussed in Remark A.9.

## 6. Discussion

**Implicit Biases.** A primary motivation arises from the observed similarity of deep layer representations in transformers and LLMs. If representations and layer contributions naturally exhibit alignment properties at convergence, gradient smoothing may offer a lens for understanding the implicit structure induced by preconditioning and a mechanism for injecting helpful inductive biases earlier in training.

**More General Operators.** Gradient Smoothing represents only one simple instantiation of the broader paradigm of Depth-wise Gradient Augmentation. While we focus on local window averaging, many other coupling operators are possible, including adaptive or learned schemes based on gradient or representation structure. The strength of coupling may also vary across depth or throughout training. More generally, while this work studies smoothing-based coupling, alternative forms of update interaction, including selective coupling or even depth-wise decoupling, may offer additional benefits. Exploring this broader design space is a promising direction for future work.

## 7. Related Work

**Transformer Layer Regularity.** Recent empirical work shows that trained transformers exhibit strong layer-wise structure in their representations and Jacobians (Aubry et al., 2025; Gromov et al., 2025; Li & Papyan, 2024; Patrawala et al., 2026; Kapl et al., 2025; Lad et al., 2025; Jiang et al., 2025b; Wolfram & Schein, 2025), with some links of such structure to improved generalization. A closely related line of work on *compression valleys* (Skean et al., 2024; 2025) shows that mid-layer parameters exhibit lower-rank structure than boundary layers, with this low-rank structure often shared across consecutive mid-layers. Such observations also motivate the exploration of layer pruning methods that

highlight minimal performance loss (Men et al., 2024; Gromov et al., 2025; Jiang et al., 2025a; Krause et al., 2025). Our work connects to this literature by showing that gradient smoothing amplifies these empirical alignment tendencies, while leading to improved performance.

**Adaptive Optimization.** Modern deep learning relies heavily on adaptive optimizers that scale updates based on gradient statistics. Adam (Kingma & Ba, 2017) and AdamW (Loshchilov & Hutter, 2019) maintain adaptive per-parameter estimates of first and second moments, enabling effective training across all modern architectures. Second-order methods such as Shampoo (Gupta et al., 2018) and SOAP (Vyas et al., 2025) approximate full-matrix preconditioning for improved convergence, while Muon (Jordan et al., 2024) applies momentum in the orthogonalized gradient space. These methods focus on adaptively scaling updates per parameter group, whereas gradient smoothing operates on a complementary axis: *depth-wise augmentations* of updates across optimization steps. Our approach is agnostic to the choice of base optimizer and can be combined with existing adaptive methods to stabilize training by amplifying gradient information across depth.

**Pre-conditioned Optimization.** Natural gradient descent (Amari, 1998) and its approximations such as K-FAC (Martens & Grosse, 2020; George et al., 2021; Eschenhagen et al., 2024; Lin et al., 2024; Nagwekar, 2025) precondition updates using curvature information to accelerate convergence. Our method adopts a pre-conditioned perspective but focuses on correlated depth structure rather than per-layer curvature, applying smoothing across transformer depth with minimal computational overhead.

**Structure in Hidden Representations.** Many works study the emergent structure of deep network representations (Wang et al., 2024a; Parker et al., 2023; Zangrando et al., 2025; Garrod & Keating, 2024; Wang et al., 2024b; Hoyt & Owen, 2021; Arous et al., 2024; Zarka et al., 2021; Ben-Shaul & Dekel, 2022; Papyan, 2020; Súkeník et al., 2023; Papyan et al., 2020; Zhou et al., 2025; Wang et al., 2026; Fisher et al., 2024), especially in transformers. In LLMs, recent works identify uniform token structures (Wu & Papyan, 2024; Shai et al., 2025; Piotrowski et al., 2025; Skean et al., 2025; 2024) and low-dimensional hidden trajectories (Song et al., 2025; Sarfati et al., 2024). Our work proposes an approach to regularize representation trajectories across depth by augmenting the gradient updates.

## 8. Conclusion

We introduced Gradient Smoothing, a simple and efficient instantiation of a broader paradigm of *Depth-Wise Gradient Augmentation*, in which optimizer updates of a layer may depend on those of other layers. The method is compatible with arbitrary base optimizers and improves training across a diverse range of settings, including LLM pretraining, RL post-training for reasoning, image classification, and diffusion modeling. Beyond performance improvements, we showed that gradient smoothing impacts the internal structure of learned representations: smoothing leads to more alignment and greater linearity of representation trajectories. We provided a theoretical characterization of this phenomenon by interpreting smoothing as a depth-wise preconditioner that contracts variation in block updates, yielding improvements in bounding the alignment of representation differences. Together, our results provide empirical evidence that leveraging block-wise structure improves both optimization and generalization in modern transformers, motivating further investigation of depth-aware optimization methods for large-scale models.

## Impact Statement

This paper presents work whose goal is to advance the field of Machine Learning. There are some potential societal consequences of improved optimization methods, including downstream impacts on systems used in science and decision-making. At the same time, the techniques introduced in this work are general-purpose and do not target any specific application domain. We do not foresee immediate negative societal impacts arising directly from this work.

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

# A. Appendix

## A.1. Specialization of Smoothing to Adam, AdamW, and Muon

In our experiments, the base optimizer $\mathcal{U}^{(t)}$ is typically Adam (Kingma & Ba, 2017) or AdamW (Loshchilov & Hutter, 2019). For completeness, we briefly recall the update rule and describe how gradient smoothing composes with it.

**Adam updates.** Let $g^{(t)} = \nabla_\theta \mathcal{L}(\theta^{(t)}, \phi^{(t)})$ denote the gradient with respect to the repeated-block parameters. Note that the the remaining parameters $\phi$ are updated in the same manner as defined below, but are just not involved in the smoothing process. Adam maintains exponential moving averages of first and second moments,

$$m^{(t)} = \beta_1 m^{(t-1)} + (1 - \beta_1)g^{(t)}, \tag{13}$$

$$v^{(t)} = \beta_2 v^{(t-1)} + (1 - \beta_2)\big(g^{(t)} \odot g^{(t)}\big), \tag{14}$$

with bias-corrected versions $\hat{m}^{(t)}$ and $\hat{v}^{(t)}$. The update direction is

$$u^{(t)} := \frac{\hat{m}^{(t)}}{\sqrt{\hat{v}^{(t)}} + \varepsilon}, \tag{15}$$

and the block parameters are updated as

$$\theta^{(t+1)} = \theta^{(t)} - \eta\, u^{(t)}. \tag{16}$$

Equivalently, Adam applies a diagonal, coordinate-wise preconditioner

$$u^{(t)} = A^{(t)} g^{(t)}, \qquad A^{(t)} := \mathrm{diag}\left(\frac{1}{\sqrt{\hat{v}^{(t)}} + \varepsilon}\right).$$

**AdamW.** AdamW applies decoupled weight decay, yielding the block-parameter update

$$\theta^{(t+1)} = \theta^{(t)} - \eta\big(u^{(t)} + \lambda\,\theta^{(t)}\big), \tag{17}$$

with non-block parameters $\phi$ updated by the base optimizer in the standard way.

**Gradient Smoothing with Adam/AdamW.** The smoothed block update is then

$$\tilde{u}^{(t)} = P\,u^{(t)}.$$

Gradient smoothing yields

$$\theta^{(t+1)} = \theta^{(t)} - \eta\,\tilde{u}^{(t)} \quad \text{(Adam)}, \tag{18}$$

$$\theta^{(t+1)} = \theta^{(t)} - \eta\big(\tilde{u}^{(t)} + \lambda\,\theta^{(t)}\big) \quad \text{(AdamW)}. \tag{19}$$

Equivalently, the effective update can be written as

$$\theta^{(t+1)} = \theta^{(t)} - \eta\,(PA^{(t)})g^{(t)},$$

highlighting that gradient smoothing composes a depth-structured preconditioner with Adam's coordinate-wise preconditioning, with $\phi$ being updated as in standard Adam/AdamW.

**Muon.** Muon (Jordan et al., 2024) treats each repeated-block parameter as a matrix and produces an approximately orthogonal update direction. It maintains a momentum buffer

$$M^{(t)} = \mu\, M^{(t-1)} + g^{(t)},$$

and forms the update direction by orthogonalizing $M^{(t)}$ via a Newton–Schulz iteration,

$$u^{(t)} := \mathrm{NS}\big(M^{(t)}\big) \approx M^{(t)}\big((M^{(t)})^\top M^{(t)}\big)^{-1/2}.$$

The block parameters are updated as $\theta^{(t+1)} = \theta^{(t)} - \eta\, u^{(t)}$. In contrast to Adam's diagonal preconditioning, $u^{(t)}$ has singular values approximately equal to one, so its per-step magnitude is largely decoupled from that of the raw gradient.

**Gradient smoothing with Muon.** The smoothed block update composes with Muon in the same way as with Adam:

$$\tilde{u}^{(t)} = P\,u^{(t)}, \qquad \theta^{(t+1)} = \theta^{(t)} - \eta\,\tilde{u}^{(t)}.$$

For matrix-valued block parameters, $P$ is applied to the orthogonalized update produced by the Newton–Schulz iteration; for non-matrix block parameters (e.g., normalization scales), to which Newton–Schulz does not apply, $P$ is applied directly to the base-optimizer update for that parameter.

## A.2. Alignment of Representation Differences under Gradient Smoothing

**Residual increments and cosine metric.** For any depth-stacked parameter vector $\vartheta = (\vartheta_1, \ldots, \vartheta_L)$, define the residual increments

$$r_\ell(\vartheta) := h_{\ell+1}(\vartheta) - h_\ell(\vartheta) = F\big(h_\ell(\vartheta); \vartheta_\ell\big) \in \mathbb{R}^d, \qquad \ell = 1, \ldots, L, \tag{20}$$

and the normalized directions

$$d_\ell(\vartheta) := \frac{r_\ell(\vartheta)}{\|r_\ell(\vartheta)\|} \in \mathbb{R}^d \quad \text{(defined whenever } r_\ell(\vartheta) \neq 0\text{)}. \tag{21}$$

Our empirical metric is

$$\cos\big(r_{\ell+1}(\vartheta), r_\ell(\vartheta)\big) = \big\langle d_{\ell+1}(\vartheta), d_\ell(\vartheta)\big\rangle, \qquad \ell = 1, \ldots, L-1. \tag{22}$$

Working with $d_\ell$ isolates *directional* alignment from magnitude effects (e.g. $\|r_{\ell+1} - r_\ell\|$ may be large even when $\cos(r_{\ell+1}, r_\ell)$ is close to 1 due to norm mismatch).

**Assumptions.** We work on a region containing the trajectories $\{h_\ell(\theta^b)\}$ and $\{h_\ell(\theta^s)\}$ and impose the following standard regularity assumptions.

**Assumption A.1.** There exist constants $L_h, L_\theta \geq 0$ such that for all $h, \bar{h} \in \mathbb{R}^d$ and $\theta, \bar{\theta} \in \mathbb{R}^p$ in the region of interest,

$$\|F(h; \theta) - F(\bar{h}; \bar{\theta})\| \leq L_h \|h - \bar{h}\| + L_\theta \|\theta - \bar{\theta}\|. \tag{23}$$

Moreover, there exist constants $0 < m \leq M < \infty$ such that, for each $\vartheta \in \{\theta^b, \theta^s\}$,

$$m \leq \min_{\ell \in [L]} \|r_\ell(\vartheta)\|, \qquad \max_{\ell \in [L]} \|r_\ell(\vartheta)\| \leq M.$$

**Lemma A.2.** *For any nonzero $a, b \in \mathbb{R}^d$,*

$$1 - \cos(a, b) = \frac{1}{2} \left\| \frac{a}{\|a\|} - \frac{b}{\|b\|} \right\|^2. \tag{24}$$

*Proof.* Expand $\|a/\|a\| - b/\|b\|\|^2 = 2 - 2\langle a/\|a\|, b/\|b\|\rangle$ and rearrange. $\square$

**Lemma A.3.** *For any nonzero $a, b \in \mathbb{R}^d$,*

$$\left\| \frac{a}{\|a\|} - \frac{b}{\|b\|} \right\| \leq \frac{2\|a - b\|}{\min(\|a\|, \|b\|)}. \tag{25}$$

*Proof.* Write

$$\frac{a}{\|a\|} - \frac{b}{\|b\|} = \frac{a - b}{\|a\|} + b\Big(\frac{1}{\|a\|} - \frac{1}{\|b\|}\Big),$$

so by the triangle inequality,

$$\left\| \frac{a}{\|a\|} - \frac{b}{\|b\|} \right\| \leq \frac{\|a - b\|}{\|a\|} + \|b\| \cdot \frac{\big|\|b\| - \|a\|\big|}{\|a\|\|b\|} \leq \frac{\|a - b\|}{\|a\|} + \frac{\|a - b\|}{\|a\|} = \frac{2\|a - b\|}{\|a\|},$$

and symmetrizing in $(a, b)$ yields (25). $\square$

**A bound on cosine misalignment.**

**Lemma A.4.** *Under Assumption A.1, for any $\vartheta \in \{\theta^b, \theta^s\}$ and any $\ell \in [L-1]$,*

$$1 - \cos\big(r_{\ell+1}(\vartheta), r_\ell(\vartheta)\big) \;\leq\; \frac{4}{m^2}\Big(L_h^2\|r_\ell(\vartheta)\|^2 + L_\theta^2\|\vartheta_{\ell+1} - \vartheta_\ell\|^2\Big). \tag{26}$$

*Consequently,*

$$\frac{1}{L-1}\sum_{\ell=1}^{L-1} \cos\big(r_{\ell+1}(\vartheta), r_\ell(\vartheta)\big) \;\geq\; 1 - \frac{4L_h^2 M^2}{m^2} - \frac{4L_\theta^2}{m^2}\cdot\frac{1}{L-1}\sum_{\ell=1}^{L-1}\|\vartheta_{\ell+1} - \vartheta_\ell\|^2. \tag{27}$$

*Proof.* Fix $\vartheta$ and $\ell$. By Lemma A.2 and Lemma A.3,

$$1 - \cos(r_{\ell+1}, r_\ell) = \frac{1}{2}\|d_{\ell+1} - d_\ell\|^2 \leq \frac{1}{2}\left(\frac{2\|r_{\ell+1} - r_\ell\|}{\min(\|r_{\ell+1}\|, \|r_\ell\|)}\right)^2 \leq \frac{2}{m^2}\|r_{\ell+1} - r_\ell\|^2.$$

Using (20) and the Lipschitz property (23),

$$\|r_{\ell+1} - r_\ell\| = \big\|F(h_{\ell+1}; \vartheta_{\ell+1}) - F(h_\ell; \vartheta_\ell)\big\| \leq L_h\|h_{\ell+1} - h_\ell\| + L_\theta\|\vartheta_{\ell+1} - \vartheta_\ell\| = L_h\|r_\ell\| + L_\theta\|\vartheta_{\ell+1} - \vartheta_\ell\|.$$

Squaring and applying $(a+b)^2 \leq 2a^2 + 2b^2$ yields

$$\|r_{\ell+1} - r_\ell\|^2 \leq 2L_h^2\|r_\ell\|^2 + 2L_\theta^2\|\vartheta_{\ell+1} - \vartheta_\ell\|^2.$$

Combining the last two inequalities gives (26). Averaging (26) over $\ell$ and using $\|r_\ell(\vartheta)\| \leq M$ gives (27). □

**Local averaging smoother and its action on update differences.** We now take $S$ to be the *local averaging* (tridiagonal) smoother with strength $\alpha \in (0, \frac{1}{2}]$:

$$(Su)_\ell = \begin{cases} \big(1 - \frac{\alpha}{2}\big)u_1 + \frac{\alpha}{2}u_2, & \ell = 1, \\ (1-\alpha)u_\ell + \frac{\alpha}{2}(u_{\ell-1} + u_{\ell+1}), & 2 \leq \ell \leq L-1, \\ \big(1 - \frac{\alpha}{2}\big)u_L + \frac{\alpha}{2}u_{L-1}, & \ell = L. \end{cases} \tag{28}$$

Let $D \in \mathbb{R}^{(L-1)\times L}$ denote the first-difference operator $(Dv)_\ell = v_{\ell+1} - v_\ell$.

**Lemma A.5** (Induced local averaging on first differences). *Let $D$ be the first-difference operator $(Du)_\ell = u_{\ell+1} - u_\ell$ for $\ell = 1, \ldots, L-1$. Let $S$ be the local averaging operator defined in (28). Define $S^{(1)} \in \mathbb{R}^{(L-1)\times(L-1)}$ by*

$$(S^{(1)}w)_\ell = \begin{cases} (1-\alpha)w_1 + \frac{\alpha}{2}w_2, & \ell = 1, \\ (1-\alpha)w_\ell + \frac{\alpha}{2}(w_{\ell-1} + w_{\ell+1}), & 2 \leq \ell \leq L-2, \\ (1-\alpha)w_{L-1} + \frac{\alpha}{2}w_{L-2}, & \ell = L-1. \end{cases}$$

*Then for all block-vectors $u$,*

$$D(Su) = S^{(1)}(Du).$$

*Proof.* Let $w := Du$, i.e. $w_\ell = u_{\ell+1} - u_\ell$. For interior $\ell \in \{2, \ldots, L-2\}$, expanding $(Su)_{\ell+1} - (Su)_\ell$ using (28) gives

$$(D(Su))_\ell = (1-\alpha)w_\ell + \frac{\alpha}{2}w_{\ell-1} + \frac{\alpha}{2}w_{\ell+1}.$$

At $\ell = 1$,

$$(D(Su))_1 = (Su)_2 - (Su)_1 = (1-\alpha)w_1 + \frac{\alpha}{2}w_2.$$

At $\ell = L-1$,

$$(D(Su))_{L-1} = (Su)_L - (Su)_{L-1} = (1-\alpha)w_{L-1} + \frac{\alpha}{2}w_{L-2}.$$

These are exactly the defining rules of $S^{(1)}$ applied to $w$, hence $D(Su) = S^{(1)}w = S^{(1)}(Du)$. □

**Lemma A.6** (Eigenvalues of induced averaging on first differences). *Let $n \geq 1$ and define $\widetilde{S}^{(n)} \in \mathbb{R}^{n \times n}$ by the tridiagonal Toeplitz form*

$$(\widetilde{S}^{(n)}w)_i = (1 - \alpha)w_i + \frac{\alpha}{2}\big(\mathbf{1}_{\{i>1\}}w_{i-1} + \mathbf{1}_{\{i<n\}}w_{i+1}\big), \qquad i = 1, \ldots, n,$$

*with $\alpha \in (0, 1)$. Then $\widetilde{S}^{(n)}$ is symmetric and has eigenpairs*

$$v_j^{(k)} = \sin\Big(\frac{\pi k j}{n + 1}\Big), \qquad \widetilde{\mu}_k^{(n)} = 1 - \alpha\Big(1 - \cos\Big(\frac{\pi k}{n + 1}\Big)\Big), \qquad k = 1, \ldots, n.$$

*In particular, $\|\widetilde{S}^{(n)}\|_{\mathrm{op}} = \widetilde{\mu}_1^{(n)} < 1$.*

*Proof.* Fix $k \in \{1, \ldots, n\}$ and set $t := \pi k/(n + 1)$. For $2 \leq j \leq n - 1$,

$$(\widetilde{S}^{(n)}v^{(k)})_j = (1 - \alpha)\sin(jt) + \frac{\alpha}{2}\big(\sin((j - 1)t) + \sin((j + 1)t)\big) = \big(1 - \alpha + \alpha \cos t\big)\sin(jt),$$

using $\sin((j - 1)t) + \sin((j + 1)t) = 2\sin(jt)\cos t$. At $j = 1$ and $j = n$ the same identity holds since the missing neighbor corresponds to $\sin(0) = \sin((n + 1)t) = 0$. Thus $v^{(k)}$ is an eigenvector with eigenvalue $1 - \alpha + \alpha \cos t = 1 - \alpha(1 - \cos t)$. The operator norm equals the largest absolute eigenvalue $\widetilde{\mu}_1^{(n)}$ since $\widetilde{S}^{(n)}$ is symmetric. $\qquad\square$

**Proposition A.7.** *Let $S^{(1)} \in \mathbb{R}^{(L-1) \times (L-1)}$ be the induced operator on differences from Lemma A.5 (i.e. with boundary diagonal $1 - \alpha$). Then for all $w \in \mathbb{R}^{L-1}$,*

$$\|S^{(1)}w\| \leq \mu_\star \|w\|, \qquad \mu_\star := 1 - \alpha\Big(1 - \cos\Big(\frac{\pi}{L}\Big)\Big) < 1, \tag{29}$$

*and hence*

$$\|w\|^2 - \|S^{(1)}w\|^2 \geq \big(1 - \mu_\star^2\big)\|w\|^2. \tag{30}$$

*Proof.* By Lemma A.6 with $n = L - 1$, the eigenvalues of $S^{(1)} = \widetilde{S}^{(L-1)}$ are $\widetilde{\mu}_k^{(L-1)} = 1 - \alpha(1 - \cos(\pi k/L))$ for $k = 1, \ldots, L - 1$. Thus $\|S^{(1)}\|_{\mathrm{op}} = \widetilde{\mu}_1^{(L-1)} = \mu_\star < 1$, giving (29). Then (30) follows from $\|S^{(1)}w\|^2 \leq \mu_\star^2\|w\|^2$. $\qquad\square$

**General one-step bound for the average cosine.** Define the average cosine across consecutive residual increments:

$$\overline{\cos}(\vartheta) := \frac{1}{L - 1}\sum_{\ell=1}^{L-1}\cos\big(r_{\ell+1}(\vartheta), r_\ell(\vartheta)\big). \tag{31}$$

We now state the general one-step comparison, allowing the current parameters to have pre-existing depth variation.

**Corollary A.8.** *Assume Assumption A.1. Let $S$ be the local-averaging smoother (28) with parameter $\alpha \in (0, \frac{1}{2}]$, and let $S^{(1)}$ be the induced $(L - 1) \times (L - 1)$ averaging operator from Lemma A.5. Define*

$$\delta := D\theta, \qquad w := Du(\theta),$$

*where $D$ is the first-difference operator and all norms and inner products are taken over depth-stacked block vectors. Then the smoothed and baseline one-step updates satisfy*

$$\overline{\cos}(\theta^s) \geq 1 - \frac{4L_h^2 M^2}{m^2} - \frac{4L_\theta^2}{m^2} \cdot \frac{1}{L - 1}\big\|\delta - \eta S^{(1)}w\big\|^2, \tag{32}$$

$$\overline{\cos}(\theta^b) \geq 1 - \frac{4L_h^2 M^2}{m^2} - \frac{4L_\theta^2}{m^2} \cdot \frac{1}{L - 1}\big\|\delta - \eta w\big\|^2. \tag{33}$$

*Moreover, the difference between the two lower bounds admits the exact identity*

$$\Big[RHS\ of\ (32)\Big] - \Big[RHS\ of\ (33)\Big] = \frac{4L_\theta^2}{m^2} \cdot \frac{1}{L - 1}\Big[\eta^2\big(\|w\|^2 - \|S^{(1)}w\|^2\big) - 2\eta\langle\delta, (I - S^{(1)})w\rangle\Big]. \tag{34}$$

*In particular, smoothing yields a strictly improved lower bound whenever*

$$\langle \delta, (I - S^{(1)})w \rangle < \frac{\eta}{2}\Big( \|w\|^2 - \|S^{(1)}w\|^2 \Big). \tag{35}$$

*A simple sufficient condition is*

$$\|\delta\| < \frac{\eta}{2} \cdot \frac{1 - \mu_\star^2}{\|I - S^{(1)}\|_{\mathrm{op}}} \|w\|, \qquad \mu_\star := \|S^{(1)}\|_{\mathrm{op}} < 1. \tag{36}$$

*Proof.* The bounds (32)–(33) follow by applying Lemma A.4 with $\vartheta = \theta^s$ and $\vartheta = \theta^b$. Indeed,

$$D\theta^s = D(\theta - \eta S u) = D\theta - \eta D(Su) = \delta - \eta S^{(1)}w,$$

where we used Lemma A.5, and similarly

$$D\theta^b = D(\theta - \eta u) = \delta - \eta w.$$

Substituting these identities into (27) gives (32) and (33).

For (34), subtract the right-hand sides of (32) and (33). The only changing term is the squared depth-difference term, and

$$\|\delta - \eta S^{(1)}w\|^2 - \|\delta - \eta w\|^2 = \eta^2 \big( \|S^{(1)}w\|^2 - \|w\|^2 \big) + 2\eta\langle \delta, (I - S^{(1)})w \rangle.$$

Multiplying by

$$-\frac{4L_\theta^2}{m^2} \cdot \frac{1}{L - 1}$$

yields (34). Condition (35) makes the right-hand side strictly positive.

Finally, (36) follows from Cauchy–Schwarz and the contraction of $S^{(1)}$. Specifically,

$$\langle \delta, (I - S^{(1)})w \rangle \leq \|\delta\| \, \|I - S^{(1)}\|_{\mathrm{op}} \|w\|,$$

while Proposition A.7 gives

$$\|w\|^2 - \|S^{(1)}w\|^2 \geq (1 - \mu_\star^2)\|w\|^2.$$

Combining these two inequalities gives the stated sufficient condition. $\qquad\square$

*Remark* A.9 (Depth-flat special case). If the current parameters are depth-flat, $\theta_1 = \cdots = \theta_L$, then $\delta = D\theta = 0$. In this case, Corollary A.8 reduces to a purely update-induced comparison:

$$\overline{\cos}(\theta^s) \geq 1 - \frac{4L_h^2 M^2}{m^2} - \frac{4L_\theta^2 \eta^2}{m^2} \cdot \frac{\|S^{(1)}w\|^2}{L - 1}. \tag{37}$$

The corresponding improvement over the baseline lower bound is

$$\Big[ \text{RHS of (37)} \Big] - \Big[ \text{same bound with } \|S^{(1)}w\|^2 \text{ replaced by } \|w\|^2 \Big] = \frac{4L_\theta^2 \eta^2}{m^2} \cdot \frac{\|w\|^2 - \|S^{(1)}w\|^2}{L - 1} \tag{38}$$

$$\geq \frac{4L_\theta^2 \eta^2}{m^2} \cdot \frac{1 - \mu_\star^2}{L - 1} \|w\|^2. \tag{39}$$

Thus, when $\eta > 0$, $L_\theta > 0$, and $w \neq 0$, depth-flat parameters yield a strict improvement in the lower bound. This special case isolates the clean mechanism: local averaging improves the bound solely by contracting depth-to-depth variation in the optimizer update. The main result, Corollary A.8, does not require this depth-flat condition and additionally accounts for interactions with pre-existing depth variation $D\theta$.

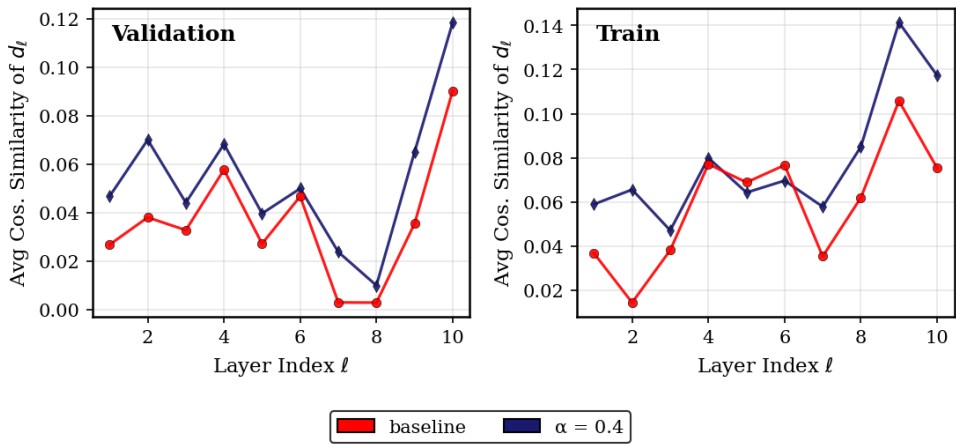

*Figure 6.* **Gradient smoothing increases layer contribution alignment.** CLS token trajectories of ViT-B trained on CIFAR-100 for 1700 epochs, comparing baseline ($\alpha = 0$, red) against heavy smoothing ($\alpha = 0.4$, navy). For each layer difference $d_\ell = x_{\ell+1} - x_\ell$, we compute the mean cosine similarity with all other layer differences $d_j$ ($j \neq \ell$). Across both validation and training sets, the smoothed model exhibits consistently higher mean cosine similarity at most layers, indicating that gradient smoothing encourages more coherent directional updates throughout the network.

### A.3. Experimental Evidence for Regularity and Smoothing

### A.4. Variance-Reduction of Gradients with Smoothing

Indeed it may be interesting to analyze the gradient update variance. Here we provide a simple variance-based analysis for the local window operator. Let $U \in \mathbb{R}^{Ld}$ denote the stacked stochastic (e.g. from minibatches) block update, and define the smoothed update by

$$\tilde{U} = (S \otimes I_d)U,$$

where $S$ is the depth-smoothing matrix.

**Proposition A.10** (Variance under window smoothing). *Let $U \in \mathbb{R}^{Ld}$ be a random vector with $\mathbb{E}\|U\|_2^2 < \infty$ and covariance $\Sigma = \mathrm{Cov}(U)$, and let $\tilde{U} = (S \otimes I_d)U$ with $S$ the local window operator (28). For any $\alpha \in [0,1]$,*

$$\mathbb{E}\|\tilde{U} - \mathbb{E}\tilde{U}\|_2^2 \leq \mathbb{E}\|U - \mathbb{E}U\|_2^2.$$

*Proof.* Since $S$ is symmetric with eigenvalues

$$\lambda_k(S) = 1 - \alpha\left(1 - \cos\frac{k\pi}{L}\right), \qquad k = 0, \ldots, L-1,$$

we have $\|S\|_2 = 1$ for $\alpha \in [0,1]$.

By linearity, $\mathrm{Cov}(\tilde{U}) = (S \otimes I_d)\,\Sigma\,(S \otimes I_d)^\top$, so the total stochastic update variance is

$$\mathbb{E}\|\tilde{U} - \mathbb{E}\tilde{U}\|_2^2 = \mathrm{tr}\big((S \otimes I_d)\,\Sigma\,(S \otimes I_d)^\top\big) = \mathrm{tr}\big(((S^\top S) \otimes I_d)\,\Sigma\big).$$

Let $A := (S^\top S) \otimes I_d$. Since $\Sigma$ is a covariance matrix, $\Sigma \succeq 0$, and clearly $A \succeq 0$ as well. Then $A \preceq \|A\|_2\, I$, which implies

$$\mathrm{tr}(A\Sigma) = \mathrm{tr}\big(\Sigma^{1/2}A\,\Sigma^{1/2}\big) \leq \|A\|_2\,\mathrm{tr}(\Sigma).$$

Since

$$\|A\|_2 = \|(S^\top S) \otimes I_d\|_2 = \|S^\top S\|_2 = \|S\|_2^2 = 1,$$

we obtain

$$\mathbb{E}\|\tilde{U} - \mathbb{E}\tilde{U}\|_2^2 \leq \mathrm{tr}(\Sigma) = \mathbb{E}\|U - \mathbb{E}U\|_2^2. \qquad \square$$

Thus, for $\alpha \in [0,1]$, window smoothing update variance is at most that of the blockwise stochastic update.

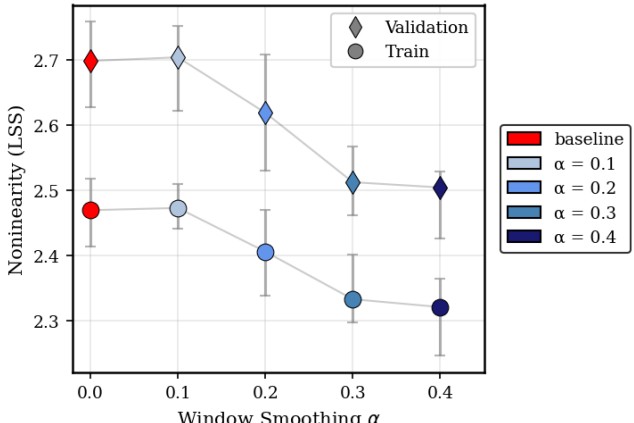

*Figure 7.* **Greater linearity with increased smoothing.** Line Shape Score (LSS) of CLS token trajectories in ViT-B trained on CIFAR-100 (1700 epochs). Each point shows the median across 100 validation (rhombus) and training (circle) images, while error bars indicate interquartile range (Q1–Q3). As the window smoothing parameter $\alpha \in \{0.1, \ldots, 0.4\}$ increases from the baseline ($\alpha = 0$), median LSS decreases monotonically, indicating that gradient smoothing produces more linear CLS token trajectories.

## A.5. Empirical Measurement of Gradient Variance

To complement the variance contraction bound of Proposition A.10, we directly measure two notions of stochastic gradient variance during pretraining and compare them across baseline and smoothing runs.

**Setup.** We train three otherwise-identical `nanochat` d24 models (1.38B parameters, 24 transformer blocks) for 7000 optimization steps: a baseline run and two window-smoothing runs with $\alpha \in \{0.05, 0.1\}$. Every 50 steps, before applying the optimizer update, we record per-microbatch gradients for each parameter group $(r, \ell)$, where $r$ indexes one of $R = 6$ roles within a block (the four attention projections and the two MLP linears) and $\ell \in \{1, \ldots, L\}$ indexes block depth. Let $g_{r,\ell,n}$ denote the flattened gradient for role $r$, layer $\ell$, and microbatch $n$, with $N$ microbatches per logging step.

**Microbatch variance.** The microbatch variance measures the across-sample noise of the stochastic gradient at each parameter group:

$$V_{r,\ell}^{\mathrm{mb}} := \frac{1}{N} \sum_{n=1}^{N} \left\| g_{r,\ell,n} - \bar{g}_{r,\ell} \right\|_2^2, \qquad \bar{g}_{r,\ell} := \frac{1}{N} \sum_{n=1}^{N} g_{r,\ell,n}.$$

Summing over roles and layers gives the total

$$V_{\mathrm{total}}^{\mathrm{mb}} := \sum_{r=1}^{R} \sum_{\ell=1}^{L} V_{r,\ell}^{\mathrm{mb}},$$

which is the empirical estimate of $\mathbb{E}\|U - \mathbb{E}U\|_2^2$ in the notation of Proposition A.10.

**Depth variance.** The depth variance measures the across-layer dispersion of the gradient within each role, evaluated per microbatch and averaged over microbatches:

$$V_r^{\mathrm{depth}} := \frac{1}{N} \sum_{n=1}^{N} \frac{1}{L} \sum_{\ell=1}^{L} \left\| g_{r,\ell,n} - \bar{g}_{r,\cdot,n} \right\|_2^2, \qquad \bar{g}_{r,\cdot,n} := \frac{1}{L} \sum_{\ell=1}^{L} g_{r,\ell,n}.$$

The total is $V_{\mathrm{total}}^{\mathrm{depth}} := \sum_r V_r^{\mathrm{depth}}$. Stacking $u = (g_{r,1,n}, \ldots, g_{r,L,n})$ along depth for a fixed role and microbatch, $V_r^{\mathrm{depth}}$ is the centered second moment of $u$ across the depth index, and so it vanishes precisely when the gradient is constant across blocks. This is the quantity that the local window operator $S$ contracts: as in Proposition A.7, $S$ shrinks depth-to-depth variation in stacked update vectors, and $V_r^{\mathrm{depth}}$ is a direct empirical proxy for this depth-roughness energy.

**Observations.** Figures 4 and 8 report $V_{\text{total}}^{\text{mb}}$ and $V_{\text{total}}^{\text{depth}}$ over training for the three runs. After the initial warmup transient (steps $\lesssim 1000$), both smoothing runs consistently exhibit lower gradient variance than baseline, in both metrics. The reduction is stable across the bulk of training and widens further in late training, consistent with Proposition A.10 acting at every step along the smoothed trajectory.

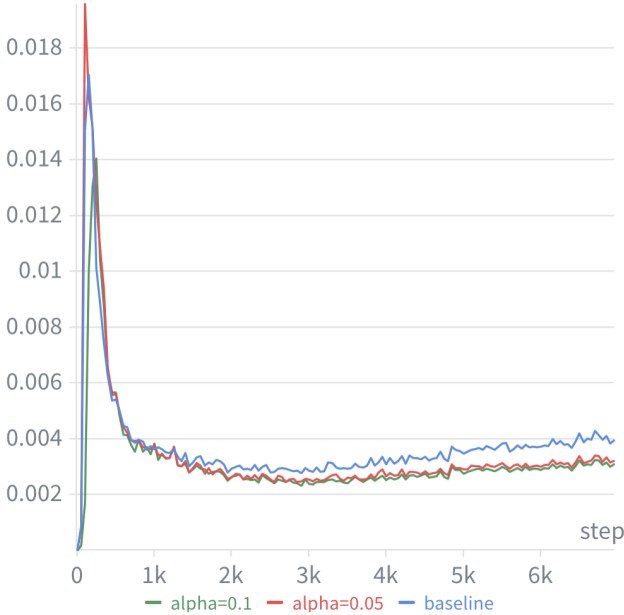

*Figure 8.* **Lower depth gradient variance with smoothing.** Total depth variance $V_{\text{total}}^{\text{depth}}$ during `nanochat` d24 pretraining (7000 optimization steps, logged every 50 steps), comparing the baseline (blue) against window smoothing with $\alpha = 0.05$ (red) and $\alpha = 0.1$ (green). Both smoothing runs show consistently lower depth-wise gradient dispersion than baseline once training leaves the warmup phase, with $\alpha = 0.05$ and $\alpha = 0.1$ producing comparable reductions.

