# OpenReview forum: "Gradient Smoothing: Coupling Layer-wise Updates for Improved Optimization"
_ICML.cc/2026/Conference — ICML 2026 regular_

### Official Review · Reviewer_6Lw2 · 2026-03-02

**Soundness:** 4
**Presentation:** 3
**Significance:** 4
**Originality:** 4
**Overall Recommendation:** 5
**Confidence:** 3

**Summary:**

Deep networks show strong representational alignment across layers during training, the authors claim that parameter updates should not be treated independently across depth.
The proposed Gradient Smoothing explicitly couples layer-wise gradient updates using simple smoothing operators across the depth dimension. It acts as a block-structured preconditioner that plugs in multiple optimizers.

**Compliance With Llm Reviewing Policy:**

Affirmed.

**Final Justification:**

Despite other reviewers point out the sufficient experimental results, I keep this score but reduce confidence.

**Key Questions For Authors:**

See Weaknesses, and:

1, Why did EMA perform better for DiTs, while Window Smoothing was preferred for LLMs and ViTs?

2, It is unclear how Gradient Smoothing interacts with the warm-up phase and learning rate decay.

**Limitations:**

The paper includes an Impact Statement, but it lacks a Limitations section.

**Strengths And Weaknesses:**

Strengths:

1, The manuscript theoretically proves that the proposed smoothing operators couple gradient updates across layers.

2, It mathematically explains the empirically observed increase in the average cosine similarity of residual increments.

3, It operates on the update vectors, requires no new trainable parameters, and incurs almost zero computational or memory overhead.

4, The authors validated it diverse architectures and training paradigms, demonstrating genuine generalization.

Weaknesses:

1, Some PEFT methods [1-3] also exploit cross-layer relationships. A brief discussion of these connections could enhance the motivation.

[1] Sharing Task-Relevant Information in Visual Prompt Tuning by Cross-Layer Dynamic Connection

[2] Visual Prompt-Agnostic Evolution

[3] CrossSpectra: Exploiting Cross-Layer Smoothness for Parameter-Efficient Fine-Tuning

2, Although the experiments cover multiple training stages, the paper does not explicitly claim this cross-stage applicability.

---

> ### Author Rebuttal · Authors · 2026-03-31
>
> We appreciate the feedback from the reviewer, and we provide our responses below.
>
> > 1 Some PEFT methods [1-3] also exploit cross-layer relationships.
>
> We thank the reviewer for suggesting these related works that highlight previous methods that leverage cross-layer relationships. Zhang et al. [3] previously exploited cross-layer relationships for low-frequency gradient updates, and are motivated by similar observations as our approach. Complementing augmentations at the gradient level, Zhou et al. [1] and Wang et al. [2] exploit similarity in token representations across depth in order to improve information propagation within models for downstream tasks. We have added additional discussion in the introduction to incorporate these changes and appreciate the relevance to our method.
>
> [1] Sharing Task-Relevant Information in Visual Prompt Tuning by Cross-Layer Dynamic Connection. Zhou et al. (2025)
>
> [2] Visual Prompt-Agnostic Evolution. Wang et al. (2026)
>
> [3] CrossSpectra: Exploiting Cross-Layer Smoothness for Parameter-Efficient Fine-Tuning. Zhang et al. (2025)
>
> > Although the experiments cover multiple training stages, the paper does not explicitly claim this cross-stage applicability.
>
> We appreciate the opportunity to improve our discussion. We now outline in detail the applicability to multiple stages of training, including our **additional results on nanochat pretraining** (response to reviewer e9K2), and including the existing GRPO fine-tuning results from the original work.
>
> > Why did EMA perform better for DiTs, while Window Smoothing was preferred for LLMs and ViTs?
>
> In response to reviewer e9K2, we have improved the diffusion baseline following Bao et al. [4]. We now feature 6.58 FID@10k, 4.01 FID@50k for the baseline and 5.72 FID@10k, 3.74 FID@50k for no LN window smoothing with $\alpha = 0.2$. In light of these additions, and in order to focus the presentation of our method, we separate the window configuration from secondary variants/ablations to the appendix. In particular, we now clarify that the paper’s core contribution is a general update-smoothing framework, with window smoothing as the main theoretically and empirically developed instance.
>
> [4] All are Worth Words: A ViT Backbone for Diffusion Models Bao et al. (2022)
>
> > 2, It is unclear how Gradient Smoothing interacts with the warm-up phase and learning rate decay.
>
> We appreciate this observation from the reviewer, and highlight that this is an interesting avenue for further ablations of our method. In our work and recent results, we avoid modifying these hyperparameters and maintain direct comparisons to baselines. Further optimizations are possible, but to focus our presentation we leave smoothing as a drop-in augmentation for a variety of existing methods. We leave further optimizations and extensions of this framework for future work.
>
> > The paper includes an Impact Statement, but it lacks a Limitations section.
>
> We have added a limitations section. Our main limitations are computational: more extensive ablations across additional domains, architectures, and hyperparameter configurations would strengthen the empirical picture but were constrained by available compute.
>
> We thank the reviewer for the opportunity to improve our discussion and for the feedback on our work.

---

> > ### Author Rebuttal · Reviewer_6Lw2 · 2026-04-03
> >
> > Despite other reviewers point out the sufficient experimental results, I keep this score but reduce confidence.

---

> > > ### Author Response · Authors · 2026-04-06
> > >
> > > We appreciate the reviewer’s continued engagement and thoughtful consideration, which have helped strengthen our work. We believe that the additional results and clarifications provided during the rebuttal period now address the remaining experimental concerns raised by the other reviewers.

---

### Official Review · Reviewer_KQnW · 2026-03-06

**Soundness:** 3
**Presentation:** 3
**Significance:** 2
**Originality:** 3
**Overall Recommendation:** 4
**Confidence:** 3

**Summary:**

This paper introduces Gradient Smoothing (GS), a training paradigm designed for optimizing deep neural networks (NNs) with repeated blocks, where the core idea is to explicitly smooth (or couple) gradient updates across different layers using structured smoothing operators. The method is motivated by the observation that modern NNs naturally develop similar representational structures across layers.  Given that standard optimizers treat each layer's parameters as independent, GS bridges this gap by applying smoothing operators such as local window averaging or exponential moving averages to the update parameters. The algorithm is generalized to various optimizers such as SGD, Adam,and AdamW.

**Compliance With Llm Reviewing Policy:**

Affirmed.

**Final Justification:**

After the rebuttal, many of my concerns were addressed. Overall, this paper presents an interesting question and a promising solution. I will stand by my positive score of 4. However, I cannot give a higher score as the current version needs more comprehensive analysis before the claims can be fully trusted. Specifically, before the authors dive into the results on how the proposed optimization methods work, it would be helpful to add a section on how "universal" the cross-layer structure is in modern NNs. Personally, I would like to see more results similar to the Qwen3-32B analysis conducted by the authors in the second round of response. This analysis does not need to be with training, but just weights and behavior analysis on existing open-sourced models across domains, including LLMs, large diffusion models etc. I believe this can significantly strengthen the quality of the paper to convince readers that gradient smoothing is necessary across the general ML community.

**Key Questions For Authors:**

- There is a phenomenon called compression valleys [1]. From my understanding, their results suggest that the mid-layer parameters exhibit a low-rank pattern while the parameters near the input and output layers are denser. This pattern suggests that even with repeated blocks, blocks with varied depth can work differently. Their results probably suggest that extreme smoothing, where every layer converges to similar parameters, might not be the ideal case, i.e., there should be a balance between smoothing and expressiveness. Could you please discuss your view on this? I wish to understand more about the connection between representational similarity among layers and compression valleys.

- This question also follows the previous one, where I wish to understand whether smoothing is detrimental in these scenarios. According to the smoothing pattern, the extreme case of smoothing would result in every layer having the same smoothed gradient, thus converging to the same parameter configuration. This is problematic, as extreme smoothing would result in each layer being indistinguishable and limited in expressiveness. What is your view on this?

 [1] "Attention Sinks and Compression Valleys in LLMs are Two Sides of the Same Coin". E. Queipo-de-Llano et al. ICLR 2026

**Limitations:**

Yes

**Strengths And Weaknesses:**

Strength:

- The paper discusses an interesting phenomenon that deep NNs usually exhibit similar representations across layers.
- The paper proposes a simple yet effective solution to inject the smooth inductive bias, i.e., applying a smoothing operator across the layers, which seems to effectively improve the model performance.

Weakness:

-  My main concern is about the empirical results. As this paper is mainly about optimization in modern models, the training/inference time dynamics could better reflect the improvement rather than the overall accuracy. For instance, the training convergence curve and efficiency (sample efficiency/number of epochs, etc.) would help in understanding the model behaviour. Also, some analysis in understanding gradient variance could be helpful.

- Also, it is unclear if this closely related representational structure across layers is general. The paper does not provide a clear discussion on the cases when the model size scales up to modern LLM level, i.e.,>30B, or even larger in the MoE ones (to the best of my knowledge, I neither find them in the previous literature). Thus, I am uncertain if this kind of pattern still exists in larger cases. It would be helpful if the authors could provide more analysis when the model size scales up.

---

> ### Author Rebuttal · Authors · 2026-03-31
>
> We appreciate the feedback; we provide our responses below.
> > 1. Training dynamics
>
> We appreciate this suggestion and agree that training dynamics are important to include.
> For the vision setting, Figure 2 already provides this perspective for ViTs by showing the validation-accuracy trajectory over training, not just the final endpoint. To further address this point, we also added language pretraining experiments under the nanochat recipe, a highly optimized setup based on the GPT-2 speedrun setting that uses NorMuon rather than Adam/AdamW. In this setting, smoothing improves both validation-loss convergence and the CORE metric for both 24-layer and 30-layer models, providing additional evidence that smoothing improves optimization dynamics in practice (https://i.postimg.cc/LsD4r1bK/image.png).
> We also emphasize that smoothing was applied directly on top of the optimized baseline recipe without additional retuning for smoothing, so the gains are not due to extra optimization effort specific to our method.
> > some analysis in understanding gradient variance could be helpful.
>
> Indeed it may be interesting to analyze the gradient update variance. Here we provide a simple variance-based analysis for the local window operator. Let $U \in \mathbb{R}^{Ld}$ denote the stacked stochastic (eg. from minibatches) block update, and define the smoothed update by
> $$
> \widetilde U = (S \otimes I_d)U,
> $$
> where $S$ is the depth-smoothing matrix. Since $S$ is symmetric with eigenvalues
> $$
> \lambda_k(S)=1-\alpha\Bigl(1-\cos\frac{k\pi}{L}\Bigr), \qquad k=0,\dots,L-1,
> $$
> we have $||S||_2=1$ for $\alpha\in[0,1]$.
>
> Let $\Sigma=\mathrm{Cov}(U)$. Then
>
> $$
> \mathrm{Cov}(\widetilde U)
> = (S\otimes I_d) \Sigma (S\otimes I_d)^\top.
> $$
>
> So, the total stochastic update variance is
>
> $$
> \mathbb{E}||\widetilde U-\mathbb{E}\widetilde U||_2^2
> = \mathrm{tr}\big((S\otimes I_d)\Sigma(S\otimes I_d)^\top\big)
> = \mathrm{tr}\big(((S^\top S)\otimes I_d)\Sigma\big).
> $$
>
> Now let $A:=(S^\top S)\otimes I_d$. Since $\Sigma$ is a covariance matrix, $\Sigma\succeq 0$, and clearly $A\succeq 0$ as well. Then
> $$
> A \preceq ||A||_2 I,
> $$
> which implies
> $$
> \mathrm{tr}(A\Sigma)
> = \mathrm{tr}(\Sigma^{1/2}A\Sigma^{1/2})
> \le ||A||_2\mathrm{tr}(\Sigma).
> $$
> Since
> $$
> ||A||_2 = ||(S^\top S)\otimes I_d||_2 = ||S^\top S||_2 = ||S||_2^2 = 1,
> $$
> we obtain
> $$
> \mathbb{E}||\widetilde U-\mathbb{E}\widetilde U||_2^2
> \le \mathrm{tr}(\Sigma)
> = \mathbb{E}||U-\mathbb{E}U||_2^2.
> $$
> Thus, for $\alpha\in[0,1]$, window smoothing update variance is at most that of the blockwise stochastic update.
>
> > 2. 30B+ models
>
> We thank the reviewer for raising this point. We agree that understanding whether the same cross-layer representational structure persists in much larger models, such as 30B+ dense LLMs or MoE models, is an important direction.
> Experiments at that scale are unfortunately beyond our current compute budget, so we will add this to the limitations section and note that scaling these results to large models is an important direction for future work.
>
> > 3. Relation with compression valleys
>
> We thank the reviewer for this connection. Compression valleys show that mid-layer parameters exhibit lower-rank structure while boundary layers remain denser, suggesting layers serve different functional roles. [1] and [2] add that this low-rank structure is often shared across consecutive mid-layers, with neighboring subspaces aligning consistently. Gradient smoothing encourages such coherence and likely has its strongest effect where alignment already occurs or is beneficial. Whether compression valleys are beneficial or detrimental, both cases motivate depth-adaptive smoothing: reinforcing coherence where helpful, or mitigating it where not. We view this as a promising extension and include further discussion in the revision and include further discussion on the relation to compression valleys in our related work.
>
> [1] Transformer Block Coupling and its Correlation with Generalization in LLMs
>
> [2] Residual Alignment: Uncovering the Mechanisms of Residual Networks
>
> > 4. Impact of extreme smoothing
>
> We thank the reviewer and agree with the observation. As with any form of regularization, smoothing becomes detrimental when applied too aggressively. Under global averaging (with a window size of the entire depth), gradients become identical and expressivity across depth collapses. Our early ablations at higher smoothing strengths (e.g., $\alpha>=0.3$) directly confirm this since in this regime we observe that performance degrades. At moderate levels, however, smoothing is consistently beneficial. Our experiments serve in part as a study of whether encouraging cross-layer coherence helps, given that it empirically occurs to some degree under standard training. We agree that excessive smoothing may suppress task-specific learning signals and reduce useful layer specialization, in analogy with other regularization methods.
>
> We greatly appreciate the feedback and opportunity to improve our work.

---

> > ### Author Rebuttal · Reviewer_KQnW · 2026-04-02
> >
> > I thank the authors for their rebuttal. Given the new visualizations and theoretical results, most of my concerns are addressed. For the larger models, just to clarify that I am not asking for any training or fine-tuning to see if the proposed methods work, but would rather see if the phenomenon that deep NNs usually exhibit similar representations across layers exists in those models. This can be done through inference-time weights analysis without large-scale experiments.
> >
> > Overall, this paper presents an interesting question and a promising solution. But I realized that there might be some comparison inconsistencies in the empirical results (from the discussion with Reviewer e9K2). Thus, I decided to maintain my score.

---

> > > ### Author Response · Authors · 2026-04-06
> > >
> > > We thank the reviewer for the follow-up and for the positive assessment of our revisions. Below we discuss the follow-up questions.
> > >
> > > **Analysis on larger models.**
> > >
> > > We appreciate the reviewer’s suggestion to examine this phenomenon at inference time in larger models. To this end, we conducted an analysis on Qwen3-32B without any additional training. Concretely, we compute both (i) the cosine similarity of the residual stream across depth, and (ii) line-shape score (LSS), on the model’s outputs on a subset of MMLU. https://postimg.cc/QH6rFXr8
> > >
> > > Our results show that both residual cosine similarity increase and linearity (smaller LSS indicates greater linearity) increase compared to random initialization, indicating that more structured and aligned representations across layers emerge as a result of training even in large-scale LLMs. This directly supports our motivating hypothesis that such cross-layer structure is not specific to small models.
> > >
> > > Additionally, we observe qualitatively different behavior in the middle layers of the trained model. This pattern is similar to the notion of compression valleys raised in the reviewer’s original feedback, notable in the pattern at mid-level layers in comparison to shallow and deep representations. We will include this analysis to further motivate the role of coupling in encouraging useful structure across depth.
> > >
> > > **Clarification on empirical comparisons.**
> > >
> > > We also thank the reviewer for pointing out the discussion with Reviewer e9K2, which led us to further clarify and strengthen our empirical evaluation.
> > >
> > > In particular:
> > > - **ViT image classification (training from scratch):** we show that our DeiT-based baselines are in line with reported ranges in the literature, and thus provide a reasonable reference point for evaluating the effect of smoothing.
> > > - **LLM reasoning (RLVR finetuning):** our baselines closely match or exceed the performance of those reported in prior work, ensuring consistency of comparison.
> > > - **Diffusion:** we replaced the original minimal rectified-flow setup with a stronger U-ViT baseline, and show that smoothing improves FID, further demonstrating that the method remains effective under a different architectural design with long skip connections.
> > > - **Language pretraining:** to further address concerns about fair baselines, we added experiments using the highly optimized nanochat setup for language pretraining. In this regime, smoothing continues to improve validation loss and CORE metric without additional hyperparameter tuning, indicating that gains are not due to extra optimization effort.
> > >
> > > Overall, we believe these clarifications and additions address the concern regarding comparison consistency and provide stronger evidence that smoothing is effective across a range of architectures, training regimes, and baseline strengths.
> > >
> > > We thank the reviewer again for their constructive feedback, which has helped improve both the empirical evaluation and the clarity of the paper, and we would appreciate any further consideration of our clarifications and updated results.

---

### Official Review · Reviewer_ry2o · 2026-03-11

**Soundness:** 3
**Presentation:** 1
**Significance:** 2
**Originality:** 3
**Overall Recommendation:** 3
**Confidence:** 4

**Summary:**

The paper presents an augmentation technique for optimization called Gradient Smoothing for training deep architectures with repeated residual layers. The proposed technique stems from the prior works’ observations on the strong directional alignment between different layers with identical topology. The authors suggest smoothing the parameter update signals by post-processing the updates by mixing with adjacent layers’ updates. Then, the paper provides experimental results on LLM, ViT, and DiT fine-tuning with improved scores and a theoretical results on layer-wise cosine similarity improvement with their method.

**Compliance With Llm Reviewing Policy:**

Affirmed.

**Final Justification:**

Even after the rebuttal, the paper presents a specific technique to Adam/AdamW optimizers rather than a truly generalizable method. I therefore still find it less promising. However, given the authors' promise to revise the manuscript, I have raised my score to weak reject, as my concerns on presentation are now resolved.

**Key Questions For Authors:**

1. Regarding Discussion section, why does improving cosine metric *beneficial* in the first place? Even though previous works report that trained residual blocks often align to a certain direction, it does not necessarily mean that the layers *should* have the identical principal directions.
2. How can the authors justify EMA version of their method theoretically?
3. Please see the weaknesses section. I believe that the current experimental results are satisfying, yet they are presented in a bit confusing way (the authors may easily improve this). However, the theoretical justification and unclear suggestion downgrades the quality of the paper. So, unless this presentation / theory soundness part is highly improved, I may not be able to give acceptance score for this version.

**Limitations:**

Yes.

**Strengths And Weaknesses:**

### Strengths

1. (Soundness) The paper presents a very simple, yet empirically effective technique called Gradient Smoothing. The shown experiments validate that the applied method can be applied to real-world fine-tuning tasks with high improvements.
2. (Significance) The paper addresses a real-world problem of fine-tuning large models. Although the initiating observation is based on the prior works’ results, the paper proves the proposed method’s efficacy through multiple experiments.
3. (Originality) The idea of averaging across different layers’ parameter update values is a simple yet novel idea, up to my knowledge.

---

### Weaknesses & Suggestion

1. (Presentation) Most significant weakness of this paper lies in its unclear connection between its proposed empirical methods, experiments, and the theoretical study in Discussion section. For example, the paper tries to suggest two distinct types of smoothing operators, local window averaging and exponential smoothing, yet the main experiments are done almost unanimously with windowed version, and EMA version is only presented for DiT training. Moreover, EMA version is not theoretically justified. Maybe the authors can just remove the EMA version completely and write a separate paper regarding that technique.
2. (Presentation) The experiments on Table 1, 2, 3, and figures are quite confusing. The hyperparameters are not systematically chosen, and the authors mix LayerNorm version with no-LayerNorm version. As a practitioner followint this work, it is less clear that which option the authors truly suggest.
3. (Significance) Although the suggested method implies to be applicable to any gradient-based optimization, the paper only proves its results on Adam and AdamW, which are basically similar to each other. There are many other well-used gradient-based optimizers, such as Prodigy or Muon, but the experiments only validate its efficacy on Adam optimizers. Adding experiments with other types of optimizers will strengthen the claim; otherwise, the paper’s claim should be focused on improving fine-tuning residual blocks with Adam.
4. (Soundness) Assumption 5.1 is way too stronger and not realistic. I am not sure why the authors use this unrealistic assumption in the first place.

---

> ### Author Rebuttal · Authors · 2026-03-31
>
> We appreciate the reviewer's feedback.
>
> W1. We agree that the presentation should better distinguish the general framework from its primary instance. The main contribution is the idea of smoothing updates across depth via a smoothing operator; local window averaging is the primary operator we study theoretically and empirically, while EMA was included only to illustrate that other filters are possible
>
> In the revision, we have made window smoothing the central method and reduced emphasis on EMA (also see our closing remarks below)
>
> W2. We agree Tables 1–3 should better separate the core setting from secondary variants. The LayerNorm distinction indicates which parameters are smoothed: all repeated-block parameters, or only nn.Linear-type (effectively excluding LayerNorm). We included this because LayerNorm parameters play a different role (feature scaling/shifting), so it is reasonable to ask whether they should be smoothed. Across all domains, we showed a simple base configuration (window smoothing, $\alpha=0.1$, non-LayerNorm parameters) already performs consistently well; nearby variants show performance can improve further. We have clarified that this is our default choice in the manuscript
>
> The hyperparameter choices may appear unsystematic, but our intent was demonstrating the paradigm works across diverse domains with simple choices (eg. $\alpha \in$ {0.1, 0.2, 0.3}), not exhaustive tuning. In the revision, we now **explicitly identify the default recommendation** and separate main results from variants illustrating flexibility
>
> W3. We appreciate this point and agree that validating the method beyond Adam/AdamW would strengthen the generality claim. To address this, **we include additional experiments on language pretraining under the nanochat recipe** [1], a highly optimized training setup that incorporates many practical improvements from the GPT-2 speedrun setting, including a variant of Muon (NorMuon) rather than Adam/AdamW, thereby demonstrating applicability beyond standard Adam-style optimizers
>
> In this setup, **we observe improvements over the baseline in both validation-loss convergence and the CORE metric**, for both 24-layer and 30-layer models **(https://i.postimg.cc/LsD4r1bK/image.png)**. Moreover, we apply smoothing directly on top of the optimized baseline recipe, without re-tuning the training setup for smoothing, so the gains are not due to additional optimization effort specific to our method; doing so could potentially improve performance even more
>
> [1] https://github.com/karpathy/nanochat
>
> W4.  We agree Assumption 5.1 is strong and unrealistic. It was included as a simple motivating step toward the more general result (without the assumption) in the appendix, but we understand this may not be clear. The **appendix already contains a more general result** not relying on Assumption 5.1 (line 783 onwards). In the revision, we now highlight this general result in the main paper and de-emphasize the assumption-based version.
>
> Q1. We agree it is not obvious a priori that improving cosine-based cross-layer similarity should always be beneficial. Our claim is not that layers should have identical principal directions, nor that maximizing such similarity is universally desirable. Rather, depth smoothing induces greater cross-layer coherence, and our experiments test whether encouraging this structure is beneficial, given that it arises naturally during standard training. In our settings, we see improvements with smoothing. As with other regularizers, we do not expect this to be monotone: excessive smoothing could suppress task-specific signals or reduce beneficial layer specialization (see also our response to Reviewer e9K2).
>
> This aligns with prior work: [2] and [3] relate smoother representation trajectories across depth to better model behavior, and [4] show stronger block coupling correlates with improved LLM performance. We have clarified this in the revision.
>
> [2] Large language models implicitly learn to straighten neural sentence trajectories to construct a predictive representation of natural language
>
> [3] Layer by Layer: Uncovering Hidden Representations in Language Models
>
> [4] Transformer Block Coupling and its Correlation with Generalization in LLMs
>
> Q2. We agree that the EMA version is not theoretically justified to the same extent as the local window operator. It was included only as a brief example that the framework is flexible and can incorporate other smoothing filters. As mentioned above, we will de-emphasize EMA by restricting to a brief appendix discussion (as suggested by the reviewer). Our main theoretical and empirical claims will remain focused on the windowed version.
>
> Q3. In the responses/revision, we have addressed these concerns: explicit practical recommendation, main results separated from variants, EMA de-emphasized, and the general theory (without Assumption 5.1) moved into the main paper. We hope these changes resolve the reviewer's remaining concerns.

---

> > ### Author Rebuttal · Reviewer_ry2o · 2026-04-03
> >
> > Thank you for the rebuttal. Regarding the presentation issue, I trust the authors' responses and will consider them as resolved. However, several concerns still remain. For instance, regarding W4, it is still unclear why Assumption 5.1 is retained despite being acknowledged as strong and unrealistic. To elevate the appendix results into the main manuscript, I believe a more substantial revision is necessary.
> >
> > Regarding W3, I expected experiments with a broader range of optimizers, if the authors want to propose techniques that can generalize beyond Adam family. The current version does not fully validate this claim. Therefore, it should be written as an upgrade of Adam-family optimizer or the authors should fully realize the generalizable capability beyond Adam/AdamW.
> >
> > In light of these points, I have adjusted my score accordingly from reject to weak reject.

---

> > > ### Author Response · Authors · 2026-04-06
> > >
> > > We thank the reviewer for the follow-up and for acknowledging the resolved presentation issues.
> > >
> > > **Regarding W4 (Assumption 5.1):**
> > >
> > > Our intent for including this was purely expository: Assumption 5.1 isolates the core mechanism of smoothing in a simplified setting, rather than serving as a necessary condition for the main result.
> > > Importantly, the appendix already proves a more general result that does not rely on this assumption (Appendix, line 783 onward). This result directly compares the smoothed and baseline one-step lower bounds on the average residual cosine, and shows that their difference consists of two components: (i) a term capturing the improvement from smoothing update differences across depth, and (ii) a term reflecting interaction with any depth variation already present in the parameters. Under depth-flat initialization (Assumption 5.1), this interaction term vanishes, recovering the simpler special case in the main text.
> > >
> > > In the revision, we will promote the more general result to the main text and remove the assumption-based version, which can be left as a remark or intermediate observation in the appendix. Since the general result is already developed in the current paper, we believe addressing this point primarily requires clarification and reorganization, rather than substantial technical revision.
> > >
> > > **Regarding W3 (generality beyond Adam-family optimizers):**
> > >
> > > We would like to clarify the scope of our claim. When we describe the framework as “general,” we mean that the **formulation is optimizer-agnostic**: smoothing operates directly on the base optimizer update vectors and can be applied on top of any base optimizer without modification. This is a property of the method’s design, not a claim that it has been exhaustively validated across all optimizers. In the original experiments, we focused primarily on Adam/AdamW because these remain the most common choices in standard training setups across the various domains we consider.
> > >
> > > That said, to empirically support applicability beyond Adam/AdamW, we added experiments during rebuttal using the nanochat training setup for language pretraining, which employs a **Muon-based optimizer (NorMuon)** rather than Adam-family methods. In this setting, which is already highly optimized, we observe consistent improvements in both validation loss and CORE metric without additional hyperparameter tuning for smoothing. This provides evidence that the method is compatible with and beneficial under a different optimizer.
> > >
> > > To avoid overstating the claim, we will revise the wording to clarify that:
> > > - the smoothing framework is **optimizer-agnostic by construction**, and
> > > - we provide empirical evidence of its benefit on both **Adam-family optimizers** and a representative **non-Adam optimizer (NorMuon)**, rather than claiming universal empirical validation across all optimizers.
> > >
> > > We again appreciate your efforts during the review process and hope these clarifications address your remaining concerns.

---

### Official Review · Reviewer_e9K2 · 2026-03-13

**Soundness:** 2
**Presentation:** 3
**Significance:** 2
**Originality:** 3
**Overall Recommendation:** 4
**Confidence:** 3

**Summary:**

The authors introduce Gradient Smoothing, a method for training neural networks, which enforces that gradients updates to each layer is smooth with respect to the layer index.

Authors claim to show that their gradient smoothing update gives generalization improvements across diverse tasks: RL finetuning of LLM, supervised learning with vision transformer, and diffusion models.

Author provide empirical evidence that gradient smoothing induces more coherent and structed representation evolution.

Authors theoretically analyze the effect of depth-wise smoothing on representational structure.

**Compliance With Llm Reviewing Policy:**

Affirmed.

**Final Justification:**

my main concerns about fairness of baseline comparisons have been addressed.

**Key Questions For Authors:**

see weaknesses

**Strengths And Weaknesses:**

## Strength
The proposition that parameter updates should be smooth with respect to neural net structure (e.g. layer index) is quite plausible.

The coupling formulation in section 3.1, based on a block-structured preconditioner, is quite clean. It nicely generalizes a few reasonable "smoothing schemes", such as local window and depth-ema, and applies to more complicated gradient-based updates such as Adam/AdamW.

## Weakness and Questions
1. Table 2 and Figure 2's reported test accuracies are quite bad, objectivey speaking, for CIFAR-100. Even a basic ResNet should achieve over 77% top-1 accuracy, which is higher than the proposed algorithm (75.62%) and the basesline (74.56%). It is possible that VIT-B is worse than CIFAR-100 than resnet. However, it is also possible that the training hyperparameters are chosen sub-optimally, making it difficult to tell if the empirical improvement is really due to depth-smoothing being superior. It is important for the authors to choose a baseline from an existing paper that was SOTA at time of publication (not necessarily sota as of today, but at least there should be some confidence that the model/training has been properly tuned.)
2. I am not familiar with whether the numbers in Table 1 for the baseline RL with AdamW are reasonable. It would be helpful if the authors can reference an existing paper to justify if the baseline numbers are consistent with what earlier papers obtained.
3. The FID number for cifar10 reported in Table 3 are quite bad, considering the authors used 50 NFE. I believe that most reasonable models in the last 3 years were able to obtain < 2 FID with < 40 NFE.

In summary, my problems with weaknesses 1-3 is not so much the authors did not get SOTA numbers, but because I am not confident that the authors were comparing to a fair baseline.

4. The discussion in Section 4.3 does demonstrate that smoothing increases cosine-similarity and LSS of successive layers. Theoretically, it is reasonable that enforcing updates to be smooth across layers causes the representation to be smooth across layers (as showin in Theorem 5.2). It is however unclear why "smooth representation across layers" is a desirable property.
5. Can the authors do a plot of "layerwise cosine similarity/LSS vs generalization error", not just for their model, but across several different tasks/models, to at least show empirical evidence that cosine/LSS correlate with improved generalization?

---

> ### Author Rebuttal · Authors · 2026-03-31
>
> We appreciate the feedback by the reviewer; we provide our responses below.
> > 1. ViT baseline
>
> For the baseline, we follow the DeiT [1] training recipe, which already includes substantial data augmentation designed to improve performance. As the reviewer notes, standard ResNets often outperform ViTs in the low-data regime, attributed to the weaker inductive biases of ViTs and their reduced data efficiency for pretraining on datasets like CIFAR100. For example, on CIFAR100, [2] report DeiT results in roughly the 72.23%–74.75% range when trained from scratch. [3] report 70.49% for DeiT-B, and [4] report results ranging from 60% to 74.51% across different ViT variants and training setups.
> We agree that our results are not state of the art in absolute terms. However, we believe the baseline is reasonable for this setting, is comparable to ViT/DeiT baselines trained from scratch in the literature, and characterizes the relative effect of depth smoothing under the hyperparameters tuned for the baseline.
>
> [1] Training data-efficient image transformers & distillation through attention
>
> [2] Adder Attention for Vision Transformer
>
> [3] Nested Hierarchical Transformer: Towards Accurate, Data-Efficient and Interpretable Visual Understanding
>
> [4] Efficient Training of Visual Transformers with Small Datasets
>
> > 2. RLVR baseline
>
> We follow the training setup of [5], and our baseline results are consistent with those reported in their work. They report AIME24: 30.0, MATH500: 83.8, and AMC23: 70.0 (while AIME25 is not reported). Our baseline closely matches on AIME24 and MATH500, and exceeds AMC23 (75 vs. 70).
>
> [5] Reinforcement Learning for Reasoning in Small LLMs: What Works and What Doesn't
>
> > 3. Diffusion baseline
>
> We thank the reviewer for identifying this and acknowledge that the original CIFAR-10 result in Table 3, based on a minimal rectified-flow implementation, is not a strong baseline for comparison against modern generative models. In response, we have added experiments using U-ViT [6], a ViT-based diffusion backbone with long skip connections proposed for image generation. Under this stronger architecture on CIFAR-10 with 50 NFE, we obtain 6.58 FID@10k, 4.01 FID@50k for the baseline and 5.82 FID@10k, 3.74 FID@50k for no LN smoothing with $\alpha = 0.2$. We believe these new results provide a substantial improvement for the evaluation of our method given rebuttal time constraints; we use identical architecture, hyperparameters, and sampling budget across all runs as the tuned baseline.
>
> [6] All are Worth Words: A ViT Backbone for Diffusion Models
> > ...not confident that the authors were comparing to a fair baseline.
>
> We appreciate this concern and agree that fair baseline comparison is critical. To address it, we add experiments on language pretraining using the nanochat recipe [7], a highly optimized setup with numerous modifications from the GPT-2 speedrun leaderboard. This provides both a strong baseline and an additional domain where smoothing remains effective. Notably, this setting uses NorMuon rather than AdamW, demonstrating applicability beyond Adam-style optimizers. We observe improvements in validation loss and CORE across both 24- and 30-layer models. (https://i.postimg.cc/LsD4r1bK/image.png)
> Also, we apply smoothing directly atop the optimized baseline without re-tuning the training recipe, so gains are not artifacts of additional optimization effort; further tuning for smoothing could yield even more gains.
> These results demonstrate smoothing's effectiveness even under strong, optimized baselines and non-Adam optimizers.
>
> [7] https://github.com/karpathy/nanochat
> > 4, 5 Why are smooth representations desirable?
>
> We thank the reviewer for this point. We agree that Theorem 5.2 and Section 4.3 alone do not show smoother cross-layer representations are universally desirable. Rather, they show depth smoothing induces a measurable structural bias toward cross-layer coherence. Our experiments empirically test whether augmenting this structure into training improves performance, and in our settings it does. We do not expect this relationship to be monotone: as with other regularizers, excessive smoothing could suppress useful task-dependent signals or reduce beneficial layer specialization.
> This view aligns with prior work on layerwise representation geometry. [8] show trained LLMs straighten representation trajectories across depth, with greater straightening linked to better next-word prediction; [9] show curvature of intermediate representations correlates with downstream performance; [10] show stronger layer coupling correlates with LLM performance. We have now clarified this in the manuscript.
>
> [8] Large language models implicitly learn to straighten neural sentence trajectories to construct a predictive representation of natural language
>
> [9] Layer by Layer: Uncovering Hidden Representations in Language Models
>
> [10] Transformer Block Coupling and its Correlation with Generalization in LLMs

---

> > ### Author Rebuttal · Reviewer_e9K2 · 2026-04-03
> >
> > I will increase my score to weak accept.

---

> > > ### Author Response · Authors · 2026-04-06
> > >
> > > We thank the reviewer for their reconsideration and for reassessing our score in response. We noticed that this change has not yet been reflected in the OpenReview system, and we would be grateful if you could update your score there when convenient.
> > >
> > > We appreciate your efforts and constructive feedback throughout the review process, which has helped us improve the clarity and quality of our paper.

---

### Decision · Program_Chairs · 2026-04-30

**Decision:**

Accept (regular)

**Comment:**

The paper proposes a new preconditioner for gradient-based methods for neural networks. The preconditioner averages the gradient across different layers and is shown to consistently improve the generalization performance in many different experiments.

Reviewers raised concerns about the CIFAR-100 experiments, presentation of the paper and theoretical results. These were addressed by the rebuttal and the remaining concerns after rebuttal are mostly a claim that the method only works for Adam/AdamW. However, authors provided convincing results on the nanochat-setup which uses Muon.  One reviewer raised their score to weak accept, but this change was not reflected in the OpenReview system.

Given that all concerns have been addressed, I recommend to accept the work in case there is room in the program. I encourage the authors to take the feedback of reviewer KQnW about universality of the cross-layer structure of neural nets into account when preparing an updated version of the manuscript.